## Report

# Meiotic cohesion requires Sirt1 and preserving its activity in aging oocytes reduces missegregation

Zihan Meng[1], Nicholas G Norwitz[1,2] & Sharon E Bickel [1 ✉]

## Abstract

Chromosome segregation errors in human oocytes increase dramatically as women age and premature loss of meiotic cohesion is one factor that contributes to a higher incidence of segregation errors in older oocytes. Here we show that knockdown of the NAD$^+$-dependent deacetylase Sirt1 during meiotic prophase in Drosophila oocytes causes premature loss of arm cohesion and chromosome segregation errors. We demonstrate that acetylation of the Sirt1 substrate H4K16 increases significantly in *sirt1* null and Sirt1 knockdown oocytes and use this as a marker for Sirt1 activity in vivo. When oocytes undergo aging, the H4K16ac signal increases significantly, consistent with an aging-dependent decline in Sirt1 deacetylase activity. However, if females are fed the Sirt1 activator SRT1720 as their oocytes age, the H4K16ac signal on oocyte DNA remains low in aged oocytes, consistent with preservation of Sirt1 activity during aging. Strikingly, age-dependent segregation errors are significantly reduced if mothers are fed SRT1720 while their oocytes age. Our data suggest that maintaining Sirt1 activity in aging oocytes may provide a viable therapeutic strategy to decrease age-dependent segregation errors.

Keywords Drosophila; Maternal Age Effect; Meiosis; Sister Chromatid Cohesion; SRT1720
Subject Categories Cell Cycle; Chromatin, Transcription & Genomics; Molecular Biology of Disease

## Introduction

Aging cells face several challenges, including (but not limited to) altered signaling pathways, metabolic changes, mitochondrial dysfunction and oxidative damage (Lopez-Otin et al, 2013; Mihalas et al, 2024). Human oocytes initiate meiosis in the fetal ovary, arrest before birth and resume meiosis upon ovulation, which can occur decades later. Therefore, in humans, oocytes undergo years of aging before they complete the first meiotic division. As women progress through their thirties, the risk that meiotic chromosome segregation errors in aging oocytes will lead to an aneuploid pregnancy increases exponentially, a phenomenon termed the maternal age effect.

Premature loss of meiotic sister chromatid cohesion is one factor that contributes to the maternal age effect (Charalambous et al, 2023; Greaney et al, 2018; Park et al, 2021; Wartosch et al, 2021). In both meiotic and mitotic cells, cohesin-mediated linkages between sister chromatids are established during S phase and are essential for accurate chromosome segregation. During meiosis, cohesion along the arms of sister chromatids stabilizes a chiasma and is required for proper segregation of homologs during anaphase I. Age-dependent loss of arm or centromeric cohesion in oocytes can lead to segregation errors during the first or second meiotic division, respectively.

The mechanisms that lead to premature loss of cohesion in aging oocytes are not well-defined. One protein that protects aging cells by regulating cellular homeostasis and stress resistance is Sirt1/Sir2 (Silent information regulator), the founding member of the highly conserved sirtuin family (Chang and Guarente, 2014; Wu et al, 2022). Originally identified as a chromatin silencing factor in yeast, Sirt1 is an NAD$^+$-dependent deacetylase with a diverse array of substrates that regulate a wide range of physiological processes including sugar and lipid metabolism, oxidative stress, inflammation, and cellular senescence (Chen et al, 2020; Grabowska et al, 2017; McBurney et al, 2013; Stunkel and Campbell, 2011; Tatone et al, 2018; Wu et al, 2022). Aging is accompanied by decreased levels of Sirt1 protein and/or activity in multiple mammalian cell types, including oocytes (Di Emidio et al, 2014; Gong et al, 2014; Ma et al, 2015; Zhang et al, 2016), and this decline is considered a key factor in health challenges that become more prevalent with aging (Chen et al, 2020; Tatone et al, 2018; Wu et al, 2022). Because of its critical role in healthy aging, Sirt1 has become a target for therapeutic interventions in the last two decades, and several small-molecule Sirt1 activators have been developed and tested with promising outcomes (Dai et al, 2018; Grabowska et al, 2017; Hubbard and Sinclair, 2014; Sinclair and Guarente, 2014; Tatone et al, 2018).

Here, we provide evidence that Sirt1 activity during meiotic prophase promotes accurate chromosome segregation in Drosophila oocytes. We show that knockdown of Sirt1 during prophase I causes premature loss of arm cohesion and missegregation of recombinant homologs during meiosis I. We previously developed an experimental procedure to age Drosophila oocytes in vivo and demonstrated that when diplotene oocytes undergo aging, the incidence of meiotic segregation errors increases significantly compared to non-aged oocytes (Perkins et al, 2019; Subramanian and Bickel, 2008). Using this oocyte aging method, we show that

[1]Department of Biological Sciences, Dartmouth College, 78 College Street, Hanover, NH 03755, USA. [2]Present address: Brigham and Women's Hospital, Harvard Medical School, Boston, MA, USA. ✉E-mail: sharon.e.bickel@dartmouth.edu

aging causes a significant increase in the H4K16ac signal on oocyte DNA, consistent with a decrease in Sirt1 activity during aging. However, if mothers are fed the Sirt1 activator, SRT1720, during the aging regimen, deacetylation of H4K16 is preserved. Furthermore, SRT1720 feeding significantly suppresses age-dependent segregation errors in *Drosophila* oocytes. Our results suggest that a nutritional supplementation strategy that preserves Sirt1 activity in aging oocytes might provide a viable therapeutic approach to attenuate the maternal age effect in humans.

# Results and discussion

## Knockdown of Sirt1 in prophase oocytes causes a significant increase in meiotic segregation errors

To determine whether accurate chromosome segregation in *Drosophila* oocytes depends on Sirt1 activity during meiotic prophase, we measured X-chromosome segregation errors in control and Sirt1 knockdown (KD) oocytes (Fig. 1A,B). The matα-GAL4-VP16 driver (hereafter matα driver) permits us to induce expression of a short hairpin exclusively in the female germline (Januschke et al, 2002). In addition, because driver expression begins in mid-prophase (Haseeb et al, 2024a; Weng et al, 2014), we can investigate the effect of Sirt1 knockdown on cohesion that was established during meiotic S phase (see Fig. 1A,B).

In our X-chromosome nondisjunction (NDJ) assay (Fig. 1D), we can recover and distinguish progeny resulting from normal or aneuploid gametes and calculate the frequency of chromosome segregation errors. We use the term NDJ broadly to include any type of segregation error.

Mild to moderate loss of arm cohesion only results in segregation errors in our NDJ assay if the oocyte achiasmate segregation system is disabled (Fig. 1C). This can be accomplished by using females that are heterozygous for a *matrimony (mtrm)* loss-of-function mutation (Harris et al, 2003). The achiasmate segregation system, which relies on heterochromatin near the centromere (yellow star in Fig. 1C), ensures that homologs that fail to attain a crossover (achiasmate homologs) still segregate accurately in Drosophila oocytes (Dernburg et al, 1996; Harris et al, 2003; Hawley et al, 1992; Karpen et al, 1996). Unfortunately, this same mechanism will mask cohesion-loss events in our NDJ assay because it also ensures accurate segregation of recombinant bivalents that lose their chiasma due to premature loss of arm cohesion (Fig. 1C).

Although we did not observe a significant difference between Sirt1 KD (1.30% NDJ) and control (0.27% NDJ, $P = 0.43$) when we tested one of the Sirt1 hairpins (SH00806.N) in $mtrm^{+/+}$ oocytes, NDJ was considerably higher when we utilized a recombinant $mtrm^{KG}$ matα driver chromosome that results in robust Gal4 expression (Perkins et al, 2019). When the genotypes in Fig. 1D were tested, X-chromosome NDJ in Sirt1 KD oocytes ($mtrm^{KG}$ matα driver → short hairpin) was significantly higher than in control oocytes ($mtrm^{KG}$ no driver → short hairpin). Two different Sirt1 hairpins yielded similar results (Fig. 1E). These findings demonstrate that Sirt1 function during meiotic prophase promotes accurate segregation in Drosophila oocytes, possibly because it helps maintain meiotic cohesion.

## Cohesion maintenance in prophase oocytes depends on Sirt1

To directly assay the state of cohesion in mature $Sirt1^{SH022-B06}$ KD and control oocytes, we performed FISH (fluorescence in situ hybridization) with oocytes that were wild-type for *mtrm*. Because the achiasmate system does not mask the cohesion-loss phenotype in a cytological assay, it does not need to be inactivated for these experiments. In addition, because recent evidence indicates that arm cohesion is weakened in $mtrm^{KG/+}$ oocytes (Bonner et al, 2020; Haseeb et al, 2024b), we performed FISH analysis using *mtrm+* oocytes so that any cohesion defects detected could be attributed solely to Sirt1 KD.

We utilized two differently labeled X-chromosome probes (Fig. 2A); one hybridizes to a large block of heterochromatin near the centromere, and the other recognizes a 100-Kb region on the distal arm. Figure 2B provides examples of oocytes in which arm cohesion is intact (one or two arm spots) and those that we score as cohesion defective (three or four arm spots). Using these same probes and FISH assay, we previously demonstrated that matα-induced knockdown of the cohesin subunit SMC3 or the cohesin loader Nipped-B causes a significant increase in the percentage of oocytes with separated arm signals, validating this method to detect cohesion defects (Haseeb et al, 2024a; Haseeb et al, 2024b).

Arm cohesion defects were significantly more prevalent in matα → $Sirt1^{SH022-B06}$ KD oocytes than in control oocytes. Figure 2C presents data from two independent replicates. Our scoring did not uncover any centromere-proximal cohesion defects in Sirt1 KD or control oocytes. However, the large size of the satellite repeat (11 Mb) recognized by this probe may hamper detection of cohesion defects near the centromere of the X chromosome (Haseeb et al, 2024a; Haseeb et al, 2024b). The finding that arm cohesion is disrupted when Sirt1 is knocked down indicates that Sirt1 function is required during meiotic prophase for cohesion maintenance in Drosophila oocytes.

## Arm cohesion loss in Sirt1 KD oocytes leads to missegregation of recombinant homologs

Sister chromatid cohesion distal to a crossover is required for chiasma maintenance, (Bickel et al, 2002; Buonomo et al, 2000; Hodges et al, 2005) and premature loss of arm cohesion in *Drosophila* oocytes will allow a recombinant bivalent to segregate randomly during meiosis I if the achiasmate system is disabled. Therefore, if Sirt1 KD causes recombinant homologs to missegregate at a higher frequency, this would provide confirmation that separated arm signals in our FISH assay are functionally relevant. Using Diplo-X progeny from the NDJ tests described above (Fig. 1E), we performed an additional cross to determine if Sirt1 KD increased missegregation of recombinant homologs (Fig. 2D).

Because the females we used in the NDJ test were heterozygous for recessive visible markers (Fig. 1D), we could deduce the X chromosome genotype of each Diplo-X female (arising from missegregation) by phenotyping her sons (Fig. 2D). Specifically, we could determine (1) whether the missegregating X chromosomes had undergone recombination before missegregation and (2) based on the centromere-proximal marker *car*, we could determine if the two chromosomes are homologs ($car^{+/-}$) or sisters ($car^{+/+}$ or $car^{-/-}$), indicating missegregation at Meiosis I or Meiosis II, respectively.

For each of the Sirt1 short hairpins for which we measured NDJ in $mtrm^{KG}/+$ oocytes (Fig. 1E), knockdown significantly increased

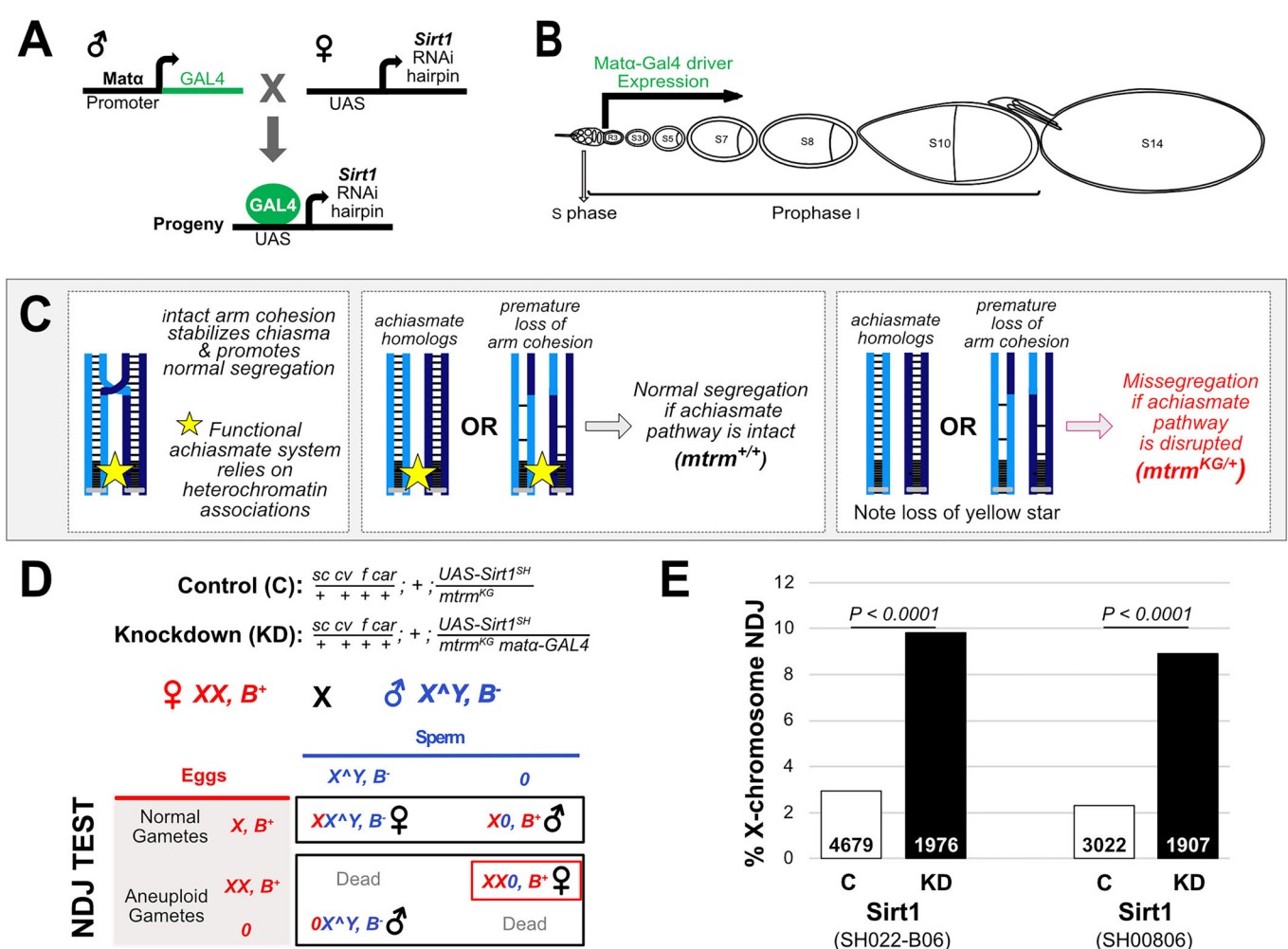

**Figure 1. Sirt1 KD in oocytes during prophase causes chromosome segregation errors.**

(A) Gal4/UAS strategy to knock down Sirt1 during meiotic prophase in the Drosophila female germline using the matα-Gal4 driver. (B) Expression of the matα driver begins 2–3 days after meiotic S phase and is active for the remainder of prophase I. (C) The achiasmate segregation system in *Drosophila* oocytes relies on heterochromatin-mediated associations to keep homologs associated. Bivalents that lose their chiasma due to loss of arm cohesion will still segregate properly if the achiasmate system is functional. Our NDJ tests utilize $mtrm^{KG}/+$ heterozygotes in which the achiasmate segregation system is disabled. (D) Genotypes for Sirt1 Knockdown and Control females used in the X-chromosome nondisjunction (NDJ) test with the $Sirt1^{SH}$ transgene on the third chromosome. Heterozygous visible markers on the X chromosome (*sc*, *cv*, *f*, and *car*) permit recombinational history analysis following the NDJ test. (E) X-chromosome NDJ is presented for two different Sirt1 short hairpins (SH022-B06 and SH00806). Black: *mtrm*, matα driver → hairpin; White: *mtrm*, no driver → hairpin). The number of flies scored (*N*) is provided in the bar for each genotype. *P* values determined as described in Zeng et al, (2010). One of two biological replicates. Source data are available online for this figure.

the frequency at which recombinant homologs missegregated (Meiosis I errors), consistent with premature loss of arm cohesion allowing chiasma destabilization prior to anaphase I (Fig. 2E). For one hairpin (SH00806), we also observed a significant increase in the frequency at which Diplo-X females inherited two sister chromatids from a crossover bivalent ($P = 0.003$) consistent with premature loss of centromeric cohesion. However, because some of our SH00806-containing stocks appeared unstable, we used only the SH022-B06 hairpin for the experiments below.

## Using the acetylation status of Sirt1 substrates on oocyte DNA to monitor Sirt1 activity in vivo

Given that reduction of Sirt1 protein and/or activity has been shown to accompany aging in multiple cell types (Di Emidio et al,

2014; Gong et al, 2014; Ma et al, 2015; Zhang et al, 2016), we next investigated whether Drosophila oocytes suffer a decline in Sirt1 protein and/or activity when they undergo aging.

We first validated immunoreagents using ovaries from $Sirt1^{SH022-B06}$ KD and control females, as well as *sirt1* null transheterozygotes ($sirt1^{4.5}/sirt1^{5.26}$). Using an anti-Sirt1 monoclonal antibody to localize Sirt1 protein in *Drosophila* ovarioles, we observed predominantly nuclear staining in nurse cells and follicle cells as well as diffuse Sirt1 signal filling the oocyte nucleus (Fig. EV1A). Nuclear enrichment of the Sirt1 signal was absent in egg chambers from *sirt1* null females (Fig. EV1B). When primary antibodies were omitted during the staining procedure (Fig. EV1D), the signal in the Sirt1 channel was similar but weaker than that observed for *sirt1* null ovaries incubated with anti-Sirt1 antibody, suggesting that non-specific antibody binding accounts for the residual signal in *sirt1* null (Fig. EV1B). Although

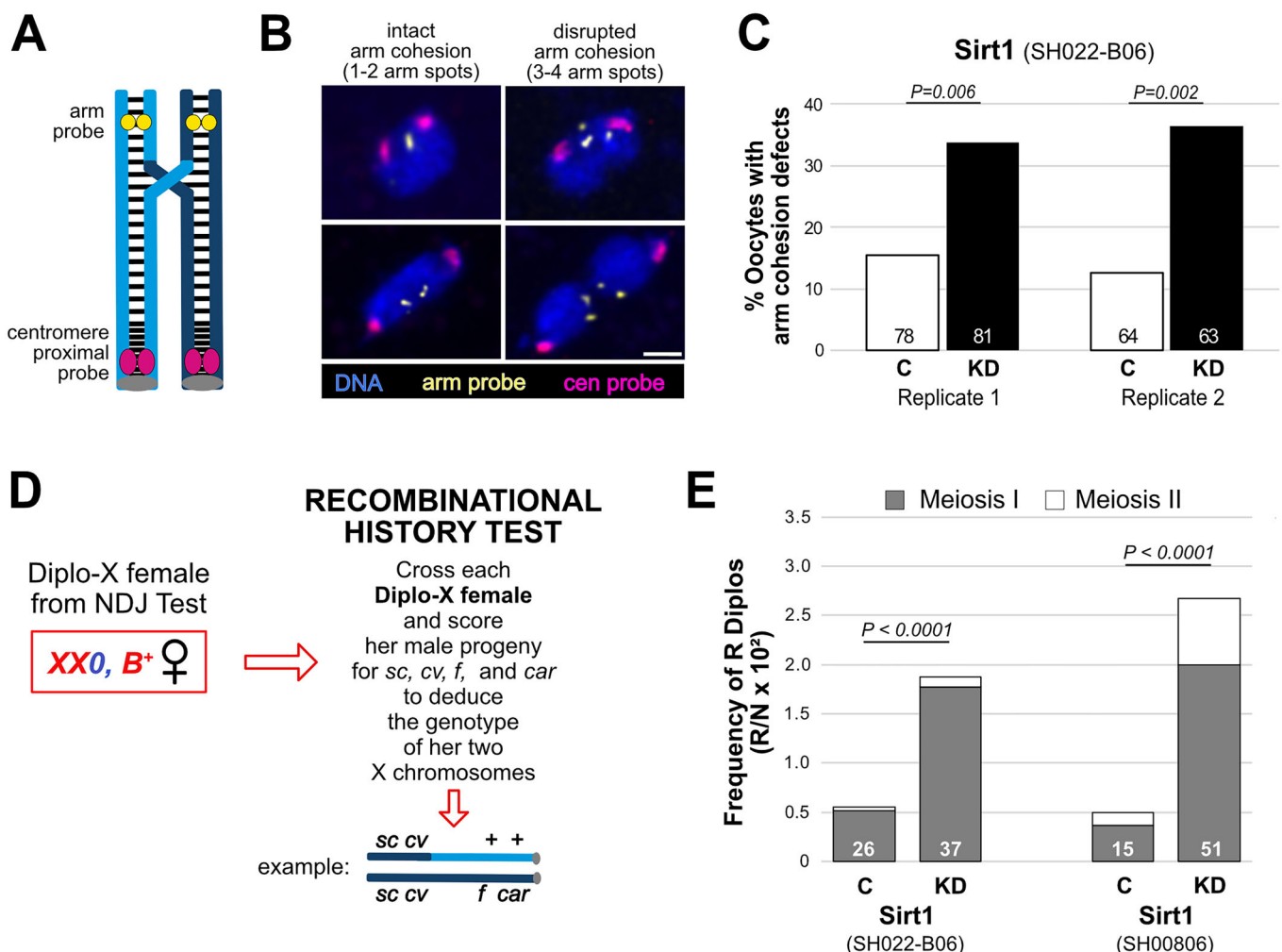

**Figure 2. Arm cohesion defects increase significantly in Sirt1 KD oocytes.**

(A) *Drosophila* X chromosome bivalent with a single crossover is shown. Light and dark blue sister chromatids are held together by cohesion, depicted as black lines. OligoPaint probe (yellow) hybridizes to a distal location on the X chromosome arm, and the 359 bp satellite repeat probe (magenta) hybridizes close to the centromere (gray). (B) Left images show oocyte chromosomes with intact arm cohesion (one or two yellow spots). Examples of premature loss of arm cohesion (three or four yellow spots) are provided on the right. Images are maximum projections of deconvolved confocal Z series. Scale bar, 2 μm. (C) The percentage of Sirt1 control (white) and KD (black) oocytes with arm cohesion defects are shown for two biological replicates. Both genotypes are *mtrm⁺*. The number of oocytes scored is indicated within each bar. *P* values were determined using a two-tailed Fisher's exact test. No centromeric cohesion defects were detected in either replicate. (D) Diplo-X females inherit both X chromosomes from their mother due to missegregation in the oocyte. By scoring the male progeny of Diplo-X females, we can determine whether a recombinant bivalent underwent missegregation and whether the two inherited X chromosomes were homologs (Meiosis I error) or sisters (Meiosis II error). (E) Diplo-X females from the NDJ tests in Fig. 1E were used to determine the recombinational history of missegregating chromosomes. The number of Diplo-X females with at least one recombinant X chromosome (R Diplo) is indicated at the bottom of each bar. The Y axis represents the number of R Diplos divided by the total number of progeny in the NDJ test (N) and multiplied by 100. Gray represents Diplo-X females that inherited two homologs (Meiosis I error), and white depicts those that inherited two sisters (Meiosis II error). P values shown compare Meiosis I errors in Sirt1 KD and control oocytes, two-tailed 2 × 2 *chi²* contingency test with Yates' correction. Source data are available online for this figure.

Sirt1 in the oocyte nucleus was greatly diminished by matα driver-induced Sirt1 KD (Fig. EV1C), the signal often remained visible in the polyploid nurse cell nuclei. As expected, the Sirt1 signal in the somatic follicle cells surrounding the egg chamber was not impacted by the germline-specific matα driver.

When we quantified the Sirt1 signal associated with oocyte DNA in control and *sirt1* null genotypes, we confirmed that Sirt1 on oocyte chromosomes was significantly reduced in null oocytes compared to control (Fig. 3A,B). Moreover, when the matα driver was used to induce knockdown of Sirt1 during meiotic prophase, the Sirt1 signal associated with oocyte chromosomes was comparable to that observed in *sirt1* null oocytes (Fig. 3A,B).

In an effort to assess Sirt1 activity in vivo, we utilized an antibody specific for histone H4 when acetylated at Lysine 16 (H4K16ac), a well-established Sirt1 deacetylation target (Shvedunova and Akhtar, 2022; Vaquero et al, 2004). As previously reported by others (Samata et al, 2020), we observed H4K16ac signal primarily on the oocyte chromosomes of *Drosophila* egg chambers with little to no staining visible in nurse cell or follicle cell nuclei (Fig. EV1). In *sirt1* null females, which lack Sirt1 enzymatic

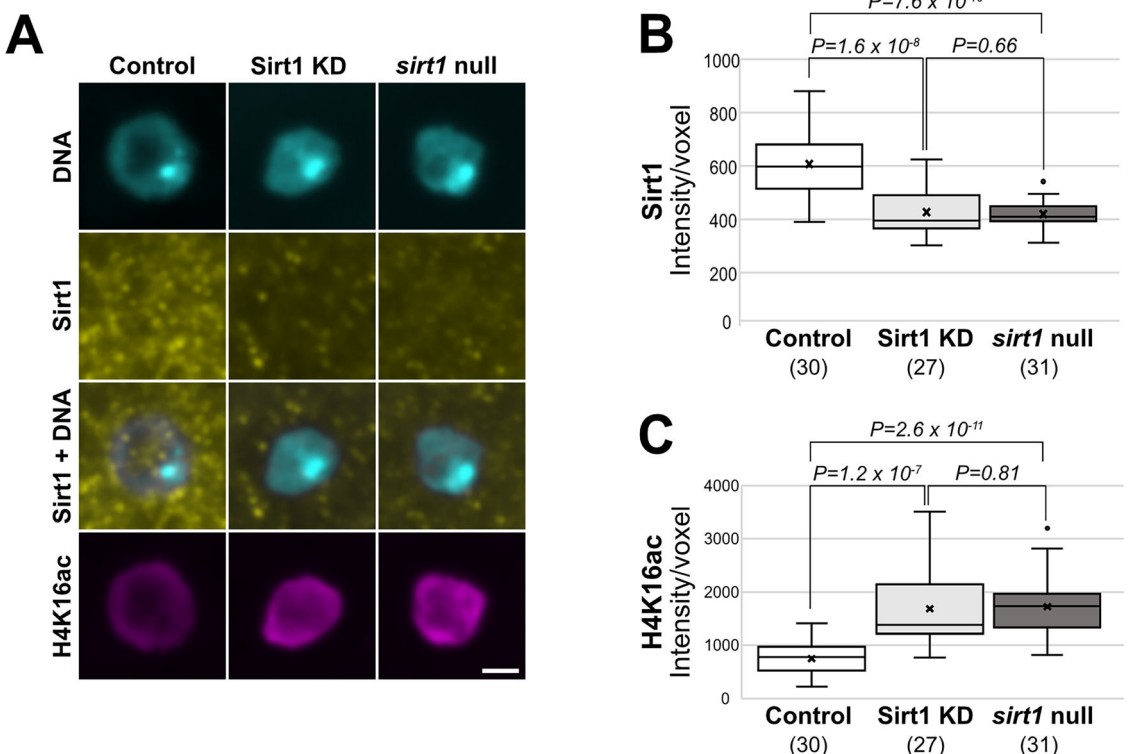

**Figure 3. H4K16ac on oocyte DNA provides a readout for Sirt1 activity in vivo.**

(**A**) Oocyte DNA (cyan) with Sirt1 (yellow) and H4K16ac (magenta) immunolocalization in Sirt1 control, Sirt1 KD and *sirt1* null oocytes (*sirt1*[4.5]/*sirt1*[5.26]). Stage 7 oocyte chromosomes are shown as a maximum intensity projection of a confocal Z series. Scale bar, 2 μm. (**B, C**) Quantification of chromosome-associated Sirt1 and H4K16ac signal intensity in stage 7-8 oocytes. The number of oocytes analyzed for each genotype is shown in parentheses. An X marks the average with horizontal lines depicting the median and quartiles. Potential outliers are denoted with a solid black dot. A two-tailed unpaired *t*-test was used to determine significance. One of two biological replicates. Source data are available online for this figure.

activity, the H4K16ac signal associated with oocyte DNA was significantly higher than in control oocytes (Fig. 3A,C). The H4K16ac signal on oocyte chromosomes was comparable in *sirt1* null and Sirt1 KD, consistent with the low level of Sirt1 protein associated with oocyte DNA in these two genotypes (Fig. 3A–C).

To further validate using H4K16ac as a marker for Sirt1 activity in vivo, we examined two additional acetylation marks. Another well-characterized Sirt1 substrate is H3K9ac (Shvedunova and Akhtar, 2022). Similar to what we observed for H4K16ac, H3K9 acetylation on oocyte DNA increased significantly in Sirt1 KD and *sirt1* null oocytes (Fig. EV2). In contrast, because H2AK9a has not been identified as a substrate of Sirt1 (Lu et al, 2018; Shvedunova and Akhtar, 2022), we used it as a negative control. The acetylation state of H2AK9 was not impacted in Sirt1 KD or *sirt1* null oocytes (Fig. EV3). Based on these data, we conclude that the H4K16ac signal on oocyte DNA provides a reliable method to monitor Sirt1 activity in the *Drosophila* oocyte.

## Aging causes acetylation of Sirt1 substrates on oocyte chromosomes to increase

We next asked whether the level of Sirt1 protein or activity associated with oocyte chromosomes is affected when *Drosophila* oocytes undergo aging. We have previously described a strategy to

age *Drosophila* diplotene oocytes in vivo using a genotype in which the achiasmate segregation system is disabled (*mtrm*[KG]/+) and functional cohesin is decreased with a deletion (*smc1Δ*/+) that halves the protein level of the cohesin subunit, SMC1 (Subramanian and Bickel, 2008). In *smc1Δ*/*mtrm*[KG] oocytes, our four-day aging regimen is sufficient to significantly increase segregation errors due to premature loss of cohesion in oocytes that arrest and age in diplotene (stages 7 and 8) (Subramanian and Bickel, 2008).

For the studies described below (cytology and NDJ), we subjected females that were heterozygous for a *mtrm*[KG] *smc1Δ* recombinant chromosome to our standard oocyte aging regimen (Fig. EV4) and focused on stages 7 and 8 for cytological analysis. As a control, we also fixed and stained ovaries from *sirt1* null females that were not subjected to the aging regimen.

Although we did not detect a decrease in the amount of Sirt1 protein associated with the DNA of oocytes that had undergone aging (Fig. 4A,B), the H4K16ac signal on oocyte DNA was significantly higher in aged oocytes than in non-aged oocytes (Fig. 4C,D). Notably, H4K16ac on the chromosomes of aged oocytes was comparable to that for *sirt1* null oocytes (Fig. 4C,D). Aging also significantly increased acetylation of H3K9, another Sirt1 substrate; however, the H2AK9ac signal did not differ between aged and non-aged oocytes (Fig. EV5). Because two well-documented Sirt1 substrates, H4K16 and H3K9, exhibit increased

acetylation when oocytes undergo aging, and a negative control behaves appropriately, we conclude that aging causes a decline in Sirt1 activity on the oocyte chromosomes.

## Feeding females SRT1720 prevents the increase in H4K16ac that accompanies oocyte aging

Several small-molecule activators of Sirt1 have been developed, and recent work suggests that nutritional supplementation may provide a valuable therapeutic approach for age-related pathologies (Dai et al, 2018; Grabowska et al, 2017; Hubbard and Sinclair, 2014; Sinclair and Guarente, 2014; Tatone et al, 2018). One of the most potent activators, SRT1720, was found to be 1000 times more effective than the naturally occurring Sirt1 activator, resveratrol

(Milne et al, 2007). In vitro, 10 µM SRT1720 was able to increase Sirt1 activity over sevenfold (Milne et al, 2007), and in vivo studies have demonstrated that feeding mice SRT1720 can delay and/or improve age-dependent health issues such as insulin resistance and inflammation (Milne et al, 2007; Minor et al, 2011; Mitchell et al, 2014). Therefore, we asked whether feeding SRT1720 to *Drosophila* females during the oocyte aging regimen could counteract the aging-induced increase in H4K16ac on oocyte chromosomes.

We carried out our standard aging regimen with two treatment groups: DMSO only and 10 µM SRT1720 dissolved in DMSO (Fig. 5A). As shown in Fig. 5B,C, aging in the absence of SRT1720 (DMSO only) caused a significant increase in the H4K16ac signal on the oocyte DNA, consistent with reduced Sirt1 activity. However, when females were fed 10 µM SRT1720 during the aging

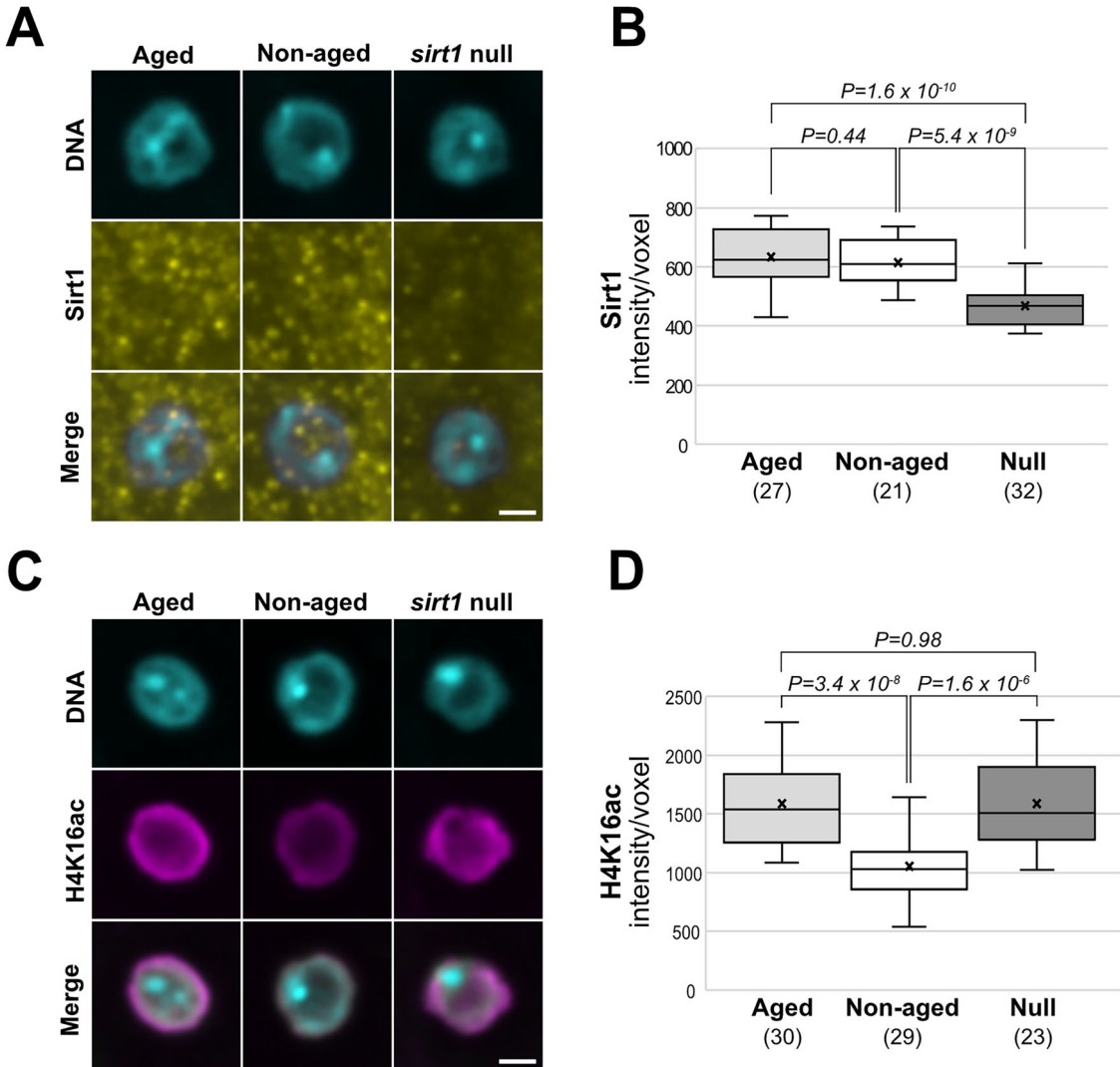

**Figure 4. Acetylation of H4K16 on oocyte chromosomes increases during oocyte aging, consistent with loss of Sirt1 activity.**

(A, C) Sirt1 (yellow) or H4K16ac immunostaining (magenta) on the DNA (cyan) of stage 7 aged, non-aged and *sirt1* null oocytes. Images are maximum intensity projections of confocal Z series. Scale bar, 2 µm. (B) Quantification of Sirt1 signal on the DNA of stage 7 and 8 oocytes. One of two biological replicates. (D) Quantification of H4K16ac signal on oocyte DNA in stage 7 and 8 oocytes. The number of oocytes scored in **B** and **D** is shown in parentheses. *P* values were determined using a two-tailed unpaired *t*-test. One of two biological replicates. (B, D) Horizontal lines depict the median and quartiles, and an X indicates the average. Source data are available online for this figure.

regimen, acetylation of H4K16 on the chromosomes of aged oocytes was comparable to the low level observed in non-aged oocytes. These data suggest that feeding females SRT1720 can prevent the decline in Sirt1 activity that occurs when their oocytes undergo aging in the absence of this Sirt1 activator.

## Age-dependent segregation errors are suppressed when mothers are fed SRT1720 during oocyte aging

Because SRT1720 feeding appeared to prevent the decline of Sirt1 activity on oocyte chromosomes during aging, we next asked

whether this supplementation could also reduce age-dependent segregation errors. We carried out the four-day oocyte aging regimen in the presence and absence of the Sirt1 activator, SRT1720, and divided the females into vials for the NDJ assay. Figure 5D and Appendix Table S1 present the data from three independent experiments. In the treatment group that lacked SRT1720 (DMSO only), meiotic segregation errors were significantly higher in aged oocytes than in non-aged oocytes, as expected. When females were fed 10 μM SRT1720 during the aging regimen, NDJ in aged oocytes was slightly higher than that observed for non-aged oocytes, but the difference between these

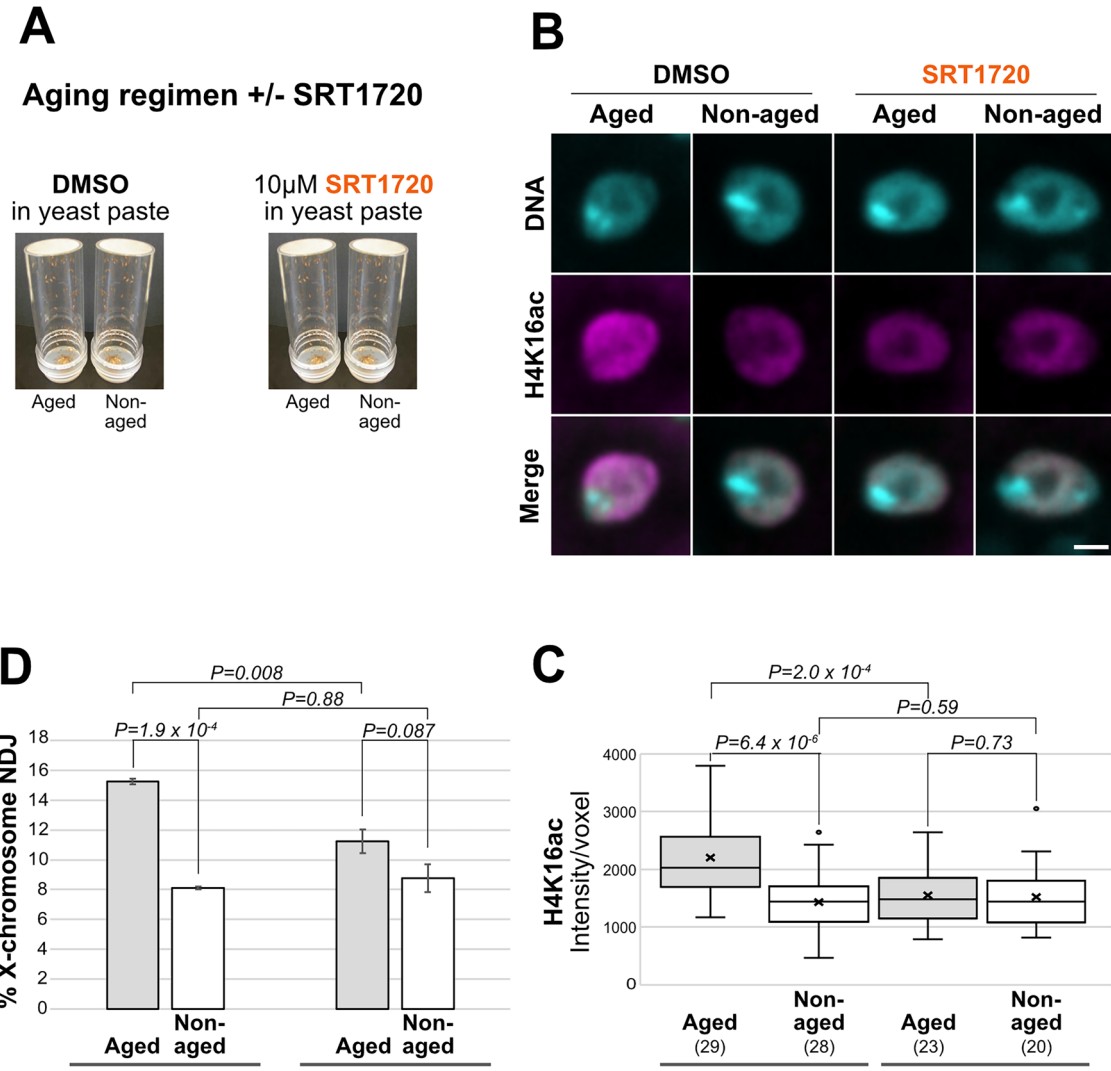

**Figure 5. Sirt1 activator feeding during oocyte aging suppresses age-dependent segregation errors.**

(A) Aging-regimen +/− SRT1720 feeding. (B) H4K16ac signal on the chromosomes of aged and non-aged oocytes (stage 7) after mothers were fed DMSO or 10 μM SRT1720 during the aging regimen. Images are maximum intensity projections of confocal Z series. Scale bar, 2 μm. (C) Quantification of H4K16ac signal intensity on oocyte DNA (stages 7 and 8) after feeding mothers DMSO or SRT1720 during the aging regimen. The number of oocytes for which the H4K16ac signal was quantified is indicated in parentheses for each treatment. An X depicts the average with horizontal lines indicating the median and quartiles. Potential outliers are denoted with a solid black dot. A two-tailed unpaired *t*-test was used to determine significance. One of two biological replicates. (D) X-chromosome segregation errors (% NDJ) were measured for aged and non-aged oocytes whose mothers were fed DMSO or SRT1720 during the aging regimen. For each condition, the mean % NDJ for three biological replicates is plotted with error bars (SEM). *P* values were calculated using a Tukey's HSD test following one-way ANOVA. Data for individual replicates provided in Appendix Table S1. Source data are available online for this figure.

two groups was not significant. Notably, segregation errors were significantly lower for aged oocytes in the SRT1720 treatment group than for aged oocytes in the DMSO treatment group. In addition, non-aged oocytes from both treatment groups (+ or – SRT1720) exhibited similar levels of NDJ. These data support the conclusion that if the aging-induced decline of Sirt1 activity on oocyte chromosomes is prevented with SRT1720 nutritional supplementation, age-dependent segregation errors can also be prevented.

## Concluding remarks

Our work has uncovered a novel role for Sirt1 during meiosis: maintenance of arm cohesion during meiotic prophase in Drosophila oocytes. Previously, a link between Sirt1 and cohesion has been described in budding yeast. Cohesion within silent chromatin domains depends on the yeast ortholog, Sir2, but does not require its deacetylase activity or its silencing partners (Wu et al, 2011). While this mechanism may operate at a limited number of other locations within the yeast genome, it is not universally required for arm cohesion in budding yeast (Chen et al, 2016). More recently, a role for Sirt1 and cohesin subunits SA1/SA2 in telomere homeostasis in human ovarian cumulus cells has been described (Valerio et al, 2018), as well as evidence that Sirt2 can deacetylate the cohesin subunit SMC1 in cultured 293 T cells (Yi et al, 2021). However, our work provides the first evidence that Sirt1 is required in oocytes to maintain arm cohesion and promote accurate chromosome segregation.

Sirt1 has a diverse set of substrates and could impact cohesion maintenance by a variety of mechanisms, either direct or indirect. Interestingly, acetylation of the cohesin loader NIPBL and the cohesion establishment factor ESCO2 are significantly elevated in Sirt1 knockout cells (Chen et al, 2012). Although these quantitative proteomics data cannot distinguish whether these proteins are deacetylated directly by Sirt1 or another deacetylase regulated by Sirt1, they suggest an intriguing mechanistic link between Sirt1 and cohesion. We have recently reported that newly synthesized cohesin loads onto oocyte chromosomes and generates cohesive linkages de novo during meiotic prophase in Drosophila oocytes, a process we have termed "cohesion rejuvenation" (Haseeb et al, 2024b; Weng et al, 2014). Moreover, Drosophila orthologs of the cohesin loader (Nipped-B) and the establishment factor (Eco) are both required for cohesion rejuvenation in Drosophila oocytes (Haseeb et al, 2024b; Weng et al, 2014). Therefore, cohesion defects in Sirt1 KD oocytes may arise because the rejuvenation process is compromised by loss of Sirt1 activity.

Sirt1 also modulates the activity of several transcription factors (Chen et al, 2020; Grabowska et al, 2017; McBurney et al, 2013; Stunkel and Campbell, 2011; Wu et al, 2022) that could impact meiotic cohesion via an indirect manner, such as controlling oxidative stress. Sirt1 positively regulates the expression or activity of several antioxidant enzymes (Alam et al, 2021; Li, 2014; Singh et al, 2018) and we have previously shown that induction of oxidative stress during meiotic prophase causes premature loss of arm cohesion in Drosophila oocytes (Perkins et al, 2016). Therefore, Sirt1 KD during meiotic prophase may lead to oxidative stress that results in premature loss of arm cohesion. Moreover, SRT1720-mediated activation of Sirt1 may slow reproductive aging and reduce segregation errors by limiting aging-induced oxidative stress in the Drosophila oocyte.

Although our 4-day aging regimen does not decrease the amount of Sirt1 protein associated with oocyte chromosomes, acetylation of two Sirt1 substrates, H4K16 and H3K9, increases significantly when Drosophila oocytes undergo aging. The robust H4K16ac signal intensity in sirt1 null oocytes and aged oocytes is comparable (Fig. 4D), suggesting that Sirt1 activity is absent in Drosophila diplotene oocytes after the aging regimen. However, if females are fed SRT1720 during the aging regimen, age-dependent chromosome segregation errors are significantly reduced, consistent with the prevention of aging-induced loss of Sirt1 activity.

If Sirt1 protein levels do not decrease, why does Sirt1 activity decline with age in Drosophila oocytes? One possibility is that $NAD^+$ decreases during aging and becomes limiting as a Sirt1 cofactor. Diminution of $NAD^+$ during aging has been reported for several tissues in mice and also in human tissues (McReynolds et al, 2020). Notably, dietary supplementation with a $NAD^+$ precursor restores oocyte fertility (Bertoldo et al, 2020) and decreases aneuploidy in chromosome spreads of meiosis II oocytes from aged mice (Miao et al, 2020). Structural and biochemical studies indicate that small-molecule Sirt1 activators bind directly to the enzyme and act allosterically to increase its affinity for NAD+ as well as substrate (Dai et al, 2015; Dai et al, 2010). Therefore, SRT1720 may suppress age-dependent NDJ errors in Drosophila oocytes because it increases the ability of Sirt1 to bind its required cofactor, $NAD^+$.

Unlike matα-induced KD of Sirt1 in Drosophila oocytes, oocyte-specific knockout of Sirt1 in mice did not lead to obvious defects in chromosome segregation (Iljas et al, 2020). However, the tools available to detect segregation errors are very different for fly (genetic) and mouse (cytological) oocytes. The timing and robust expression of the matα driver has been critical for our interrogation of cohesion maintenance after normal establishment occurs in S phase. In addition, lower redundancy in the Drosophila genome than in mice may make it easier to uncover a phenotype caused by loss of Sirt1 function (Bier, 2005; Obafemi et al, 2025; Verheyen, 2022). While mammals have seven sirtuins, Drosophila only has five, and lacks a clear Sirt2 ortholog (Rahman et al, 2014). Finally, evidence of a backup system that promotes accurate segregation of achiasmate chromosomes has been reported for yeast meiotic cells and mouse spermatocytes (Kurdzo and Dawson, 2015) and proposed for human oocytes (Koehler and Hassold, 1998). If such a system does exist in mammalian oocytes, it may prevent missegregation even if bivalents suffer cohesion loss due to oocyte-specific knockout of mouse Sirt1.

While we acknowledge that using Drosophila as a model for aging human oocytes has limitations, we consider this system a valuable tool to better understand the mechanisms underlying the maternal age effect. Our age-dependent NDJ assay (using $mtrm^{KG}$ $smc1\Delta/+$ heterozygotes) is sensitized for the detection of segregation errors arising from premature loss of cohesion. Although we limit our aging regimen to 4 days for technical reasons, stage 7 and 8 oocytes spend 11-18X more time in diplotene when they arrest and age (Subramanian and Bickel, 2008). Furthermore, only diplotene oocytes are vulnerable to age-dependent NDJ; segregation errors are not elevated in oocytes that arrest and age prior to synaptonemal complex disassembly (Subramanian and Bickel, 2008).

In conclusion, our data strongly support the model that preserving Sirt1 function during aging prevents age-dependent chromosome segregation errors in *Drosophila* oocytes. We hope these findings will inform further exploration of Sirt1 activation as a mechanism to preserve the fidelity of chromosome segregation as oocytes age.

# Methods

## Reagents and tools table

| Reagent/resource | Reference or source | Identifier or catalog number |
| --- | --- | --- |
| **Experimental models** | | |
| See Appendix Table S2 for *D. melanogaster* stocks used in this study | This study | N/A |
| **Antibodies** | | |
| Polyclonal rabbit anti-acetyl-H4K16 | Millipore | Cat# 07-329; RRID: AB_310525 |
| Polyclonal rabbit anti-acetyl-H3K9 | Abcam | Cat# 10812 RRID: AB_297491 |
| Polyclonal rabbit anti-acetyl-H2AK9 (serum) | Active Motif | Cat# 39109 RRID: AB_2793158 |
| Monoclonal mouse anti-Sirt1, clone p4A10 | Developmental Studies Hybridoma Bank, depositor S. Parkhurst | Clone P4A10; RRID: AB_1553778 |
| Purified p4A10 mouse anti-Sirt1 antibody from p4A10 clone | Bio X Cell, Lebanon, NH | |
| Cy3 Donkey anti-rabbit | Jackson ImmunoResearch | Cat# 711-165-152; RRID: AB_2307443 |
| Cy5 Donkey anti-rabbit | Jackson ImmunoResearch | Cat# 711-175-152; RRID: AB_2340607 |
| Cy5 Donkey anti-mouse | Jackson ImmunoResearch | Cat# 715-175-151; RRID: AB_2340820 |
| **Oligonucleotides and sequence-based reagents** | | |
| Alexa 647-labeled Oligopaint probe (OPP122), Mixture of 80-base oligos targeting 100 kb distal region of the X chromosome (dm6, nucleotides 1,400,000–1,500,000) | Joyce Lab, University of Pennsylvania | N/A |
| Cy3-conjugated probe (50 -Cy3-AGGGATCGTTAGCACTCGTAAT) hybridizes to the 359-bp repeat in centromere-proximal heterochromatin of the X chromosome | Integrated Technologies | N/A |
| **Chemicals, enzymes and other reagents** | | |
| InSolution SRT1720, HCl (25 mM in DMSO) | Millipore | Cat# 530748 |
| Filter-sterile dimethyl sulfoxide (DMSO) | Millipore | Cat# D2438 |
| Active dry yeast | Red Star | |
| Agar powder/flakes | Fisher Scientific | Cat# BP1423-500 |
| Bovine Serum Albumin | Fisher Scientific | Cat# BP1605-100 |
| DAPI | Invitrogen | Cat# D1306 |
| Formaldehyde, 16% | Ted Pella | Cat# 18505 |
| Formamide | Invitrogen | Cat# AM9342 |
| Grace's Medium | Thermo Fisher | Cat# 11595030 |
| Heptane | Fisher Scientific | Cat# H-350-4 |
| Hoechst 33342 | Thermo Fisher | Cat# H3570 |
| Normal donkey serum | Jackson ImmunoResearch | Cat# 017-000-121 |
| Poly-L-lysine | Sigma-Aldrich | Cat# P8920 |
| RNase A (10 mg/mL) | Thermo Fisher | Cat# EN0531 |
| 10% Tween-20 | Thermo Fisher | Cat# 28320 |
| 10% Triton X-100 | Thermo Fisher | Cat# 28314 |
| Prolong Gold Antifade | Thermo Fisher | Cat# P36930 |
| SlowFade Diamond Antifade | Thermo Fisher | Cat# S36967 |
| **Software** | | |
| Volocity visualization, restoration, and quantitation | Version 6.5.0 https://www.volocity4d.com/download | |
| Nikon elements (for spinning disc confocal imaging) | Version 5.11.02 Build 1369 | |
| MATLAB(MathWorks) | Version R2022a https://www.mathworks.com/products/new_products/release2022a.html | |
| Affinity designer | Version 1.10.6.1665 https://affinity.serif.com/en-us/designer/ | |
| Microsoft Office | Version 16.84 | |
| Statistics Kingdom online calculator (One-way ANOVA calculator and Tukey HSD) | https://www.statskingdom.com/180Anova1way.html | |

## Fly stocks and crosses

Stocks and crosses were raised on standard cornmeal-molasses food and kept at 25 °C in a humidified incubator. Appendix Table S2 provides full genotypes for all stocks utilized in this study as well as Bickel stock numbers, which accompany the cross descriptions below. All fly work was reviewed and approved by the Institutional Biological Safety Committee at Dartmouth.

## X-chromosome NDJ and recombinational history assays

To measure chromosome segregation errors in control and Sirt1 KD oocytes, *y sc cv v f car;+; mtrm^{KG}/TM3,Sb* (M-835) or *y sc cv v f car;+; mtrm^{KG}, matα/TM3,Sb* (M-834) males were crossed to *y* virgins containing a UAS-Sirt1 short hairpin (SH) insertion on the 2^{nd} chromosome (*Sirt1^{SH022-B06}*, H-087) or the 3^{rd} chromosome (*Sirt1^{SH00806}*, H-084). Non-balancer **Control** (*mtrm^{KG}*, no driver → *Sirt1^{SH}*) and **KD** (*mtrm^{KG}*, matα driver → *Sirt1^{SH}*) female progeny were collected as virgins and mated to *X^Y, Bar* (C-200) males in vials containing food and a small amount of dry yeast. For each resulting Sirt1 control and KD genotype, 20 vials of the NDJ cross (8 virgins × 4 males) were set on day 0, the parents cleared on day 7, and the progeny scored daily from day 11 through day 18. Because all the progeny from normal gametes but only half of the progeny from exceptional gametes survive (Fig. 1D), % NDJ is calculated using the following formula: $[(2(\text{Diplo-X} + \text{Nullo-X}))/(N + 2(\text{Diplo-X} + \text{Nullo-X}))]100$, where $N$ is the total number of progeny scored. To determine whether Sirt1 KD significantly impacted the fidelity of meiotic chromosome segregation, $P$ values were calculated for each hairpin by comparing control and KD data using the calculator developed by Gilliland and colleagues (Zeng et al, 2010).

The X-chromosome genotype of the females used for the NDJ assay (*y/ y sc cv v f car*) allowed us to perform an additional test to determine the recombinational history of the missegregating X chromosomes inherited by the Diplo-X progeny (Fig. 2D). When scoring the NDJ tests, Diplo-X progeny were collected each day and phenotyped for the X-chromosome visible markers *sc, cv, f*, and *car*. Each female was mated to two *y w* (A-062) males, and the parents cleared on day 7. Male progeny were scored for *sc, cv, f*, and *car* through day 18. By considering the X-chromosome phenotype of the Diplo-X female and her sons, one may deduce the genotype of the two X chromosomes that missegregated, including whether they recombined before missegregation and whether they were sisters (*car^{+/+}* or *car^{-/-}*) or homologs (*car^{+/-}*). The frequency at which recombinant X chromosomes underwent missegregation in each genotype was calculated by dividing the number of Diplo-X females that inherited at least one recombinant chromosome by the total number of progeny scored in the NDJ test and multiplying by 100. Meiosis I and Meiosis II errors were graphed individually and based on whether the Diplo-X female inherited two homologs (Meiosis I) or two sisters (Meiosis II). A two-tailed $2 \times 2$ *chi²* contingency test with Yates' correction (GraphPad) was used to determine whether the frequency of Diplo-X females with at least one recombinant X chromosome was significantly different between Sirt1 KD and control. $P$ value calculations were performed separately for Diplo-X females that inherited two homologs (Meiosis I) and Diplo-X females that inherited two sisters (Meiosis II).

Not all Diplo-X females were fertile, and a small number of Diplo-X genotypes could not be confidently called. This assay will underestimate the number of recombinant bivalents because only two of the four chromatids will be inherited by the Diplo-X female (and therefore scored) and double crossovers within the large interval between *cv* and *f* will be invisible. In addition, although crossovers are unlikely within the 3.5 cM interval between *car* and heterochromatin, a small number may still occur.

## FISH to score for cohesion defects

We utilized two X chromosome probes so that we could score for arm and centromeric cohesion defects for the same chromosome assayed in the NDJ and Recombinational History tests. The Alexa 647-labeled OligoPaint probe (OPP122) contains a mixture of 80-mers that recognize a 100-kb distal region on the X chromosome and allows us to assess the state of arm cohesion. The Cy3-labeled pericentromeric probe hybridizes to an 11 Mb stretch of satellite DNA near the centromere of the X chromosome. Unfortunately, the large size of this target may limit our detection of cohesion defects near the X chromosome centromere (Haseeb et al, 2024a; Haseeb et al, 2024b).

To generate control and KD genotypes, *y w* (A-062) or *matα* (T-273) males were crossed to *Sirt1^{SH022-B06}* virgins (H-213). About 20–25 young female progeny were held with ten males in a vial with food and dry yeast for 3 days before dissection.

Below is a brief description of the steps used in FISH detection of sister chromatid cohesion defects in mature Drosophila oocytes (stages 13–14). Additional details, especially regarding removal of chorions and vitelline membranes and tips for oocyte handling, may be found in (Perkins and Bickel, 2017). A detailed protocol is available upon request.

In a shallow dissecting dish, 15–20 sets of ovaries were dissected within a 10 min window in 1X Modified Robb's buffer (55 mM potassium acetate, 40 mM sodium acetate, 100 mM sucrose, 10 mM glucose, 1.2 mM magnesium chloride, 1.0 mM calcium chloride, 100 mM Hepes, pH 7.4) and transferred to a 1.5 mL microfuge tube. After the addition of 500 μL prewarmed (37 °C) fixative (4% formaldehyde, 100 mM sodium cacodylate, 100 mM sucrose, 40 mM sodium acetate, 40 mM sodium acetate, 10 mM EGTA) and 500 μL of heptane (room temperature, RT), the tube was mixed vigorously and the ovaries were fixed for 6 min on a nutating mixer. Following removal of fixative, ovaries were washed three times with 500 μL of 1X PBSBTx (1X PBS/0.5% BSA/0.1% Triton X-10) by inverting the tube two to three times and allowing the ovaries to settle. Fixed ovaries were transferred to a shallow dissecting dish containing PBSBTx and pipetted up and down using a BSA-coated gel-loading tip to help dissociate the later stages. This mixture was rolled between the frosted regions of two slides to manually remove the chorion and vitelline membrane of mature oocytes and transferred to a 15 mL conical tube for three rounds of settling (3 mL of PBSBTx) to remove younger stages and debris, which settle less quickly. Mature oocytes were transferred to a 500 μL microfuge tube and stored in PBSBTx overnight at 4 °C. The next morning, oocytes were rinsed once with 500 μL PBSTx (1X PBS containing 1% Triton X-100) and incubated on a nutating mixer for 2 h at RT in PBSTx containing 100 μg/mL RNAse. Following three rinses in 2X SSCT (0.3 M sodium chloride, 30 mM sodium citrate with 0.1% Tween-20), oocytes were washed three times 10 min in 2X SSCT, followed by a 10 min wash in 2X SSCT containing 20% formamide, another with 2X SSCT / 40% formamide and a final wash in 2X SSCT/50% formamide, all at RT on a nutating mixer. Oocytes were incubated in 2X SSCT + 50% formamide with rotation for 2 h at 37 °C before transfer to 200 μL PCR tubes and a pre-denaturation program in a thermocycler (5 min @ 37 °C, 3 min @ 92 °C, 20 min @ 60 °C, Hold @ 37 °C). Probes in 50 μL of 1X hybridization buffer (3X SSC, 50%

formamide, 10% dextran sulfate) were allowed to hybridize overnight after denaturation (37 °C for 5 min, 92 °C for 3 min, Hold @ 37 °C). The Cy3-labeled centromere-proximal probe was used at 1 ng/µL, and the Alexa 647-labeled Oligopaint probe (OPP122) was used at 0.50 pmol/µL. On day 3, oocytes were transferred to a prewarmed 500 µL microfuge tube and taken through a series of 500 µL washes at 37 °C on a rotator: three 20 min washes in 2X SSCT/50% formamide, three 10 min washes in 2X SSCT/50% formamide. Washes continued on a nutating mixer: one 10 min wash in 2X SSCT/40% formamide, one 10-min wash in 2X SSCT/20% formamide and one 10 min wash in 2X SSCT. Following a 30 min incubation with DAPI (1 µg/mL) in 2X SSCT (protected from light on a nutating mixer), oocytes were rinsed three times with 2X SSCT followed by two 10 min washes in 2X SSCT. Oocytes were rinsed with 1X PBS/0.01% Tween-20 and left in that solution for mounting onto poly-L-lysine-coated 18 mm #1.5 coverslips with 25 µL of Prolong Gold mounting medium. Slides were allowed to cure in the dark for at least 21 days before imaging.

## Sirt1, H4K16ac, H3K9ac, and H2AK9 Immunostaining

matα (T-273) or y w (A-062) males were crossed to Sirt1$^{SH022-B06}$ virgins (H-213) to produce Sirt1 control and KD ovaries/oocytes. To generate sirt1 null oocytes, sirt1$^{5.26}$ males (B-193) were crossed to sirt1$^{4.5}$ females (B-194) and the ovaries of female progeny were examined. Each of these sirt1 deletion alleles removes >750 nucleotides of the Sirt1 coding sequence but leaves the coding sequence of the adjacent DnaJ-H gene intact (Newman et al, 2002). For the above genotypes, newly eclosed females were held with males for 2–3 days in vials with food and a small amount of dry yeast before ovary dissection

For immunostaining aged and non-aged ovarioles, mtrm$^{KG}$ smc1Δ / TM3 males (M-822) were crossed to y w females (A-062). Non-balancer virgins from this cross were collected during an 8–12 h interval, held overnight in vials with food and a small amount of dry yeast, and subjected to the 4-day aging regimen (Fig. EV4, and described in more detail below). Ovary dissections were performed immediately following the aging regimen.

For each genotype or treatment condition, six sets of ovaries were dissected in Grace's medium, and ovarioles were gently splayed (early stages) using a fine tungsten needle.

Fixation was performed in a deep-well glass dish with gentle rotation on a shaker for 20 min at RT in 400 µL of 1X PBS containing 2% formaldehyde. All subsequent incubations and washes were performed in a glass dish at RT with gentle rotation on a shaker. Following fixation, ovaries were rinsed three times with 400 µL of 1X PBS/0.2% Triton X-100 and permeabilized by performing two 15-min incubations in 400 µL of 1X PBS containing 0.5% Triton X-100. After three rinses with 400 µL of 1X PBS/0.2% Tween-20, ovaries were blocked for 1 h in 400 µL of blocking buffer (1X PBS/0.2% Tween-20/0.5% BSA/5% Donkey Serum). Ovaries were incubated overnight in a humidified box in 200 µL of 1X PBS/0.01% Tween-20/0.5% BSA containing primary antibody. The next morning, 400 µL of 1X PBS / 0.2% Tween-20 was used for each of three rinses, followed by three 20 min washes. Ovaries were incubated for 1 h, protected from light, in 200 µL in 1X PBS/0.01% Tween-20/0.5% BSA containing secondary antibody. Following three rinses and a 20 min wash with 400 µL of 1X PBS/0.2%

Tween-20, ovaries were incubated 20 min in 1X PBS containing Hoechst (2.0 µg/mL), and 20 min in 1X PBS/0.01% Tween-20. After separation with fine tungsten needles, ovarioles were transferred onto 18 mm poly-L-lysine-coated #1.5 coverslips, excess liquid removed, and a slide with mounting medium lowered onto the coverslip.

Either SlowFade Diamond (Figs. 3 and 4) or ProLong Gold (Fig. 5) was used for mounting. SlowFade Diamond provided the highest signal-to-noise ratio for anti-Sirt1 staining, but the signal did degrade over the course of a week, so all images were captured within three days of mounting. Coverslips were sealed with nail polish immediately upon SlowFade mounting, and slides were stored flat and protected from light at 4 °C. For ProLong Gold, slides were allowed to cure in the dark for at least 14 days before application of nail polish and imaging.

Purified anti-Sirt1 antibody (~52 mg at 3.7 mg/mL) was produced by Bio X Cell (Lebanon, NH) using the mouse hybridoma cell line P4A10 that we obtained from the Developmental Studies Hybridoma Bank (DSHB). Mouse anti-Sirt1 antibody was used at 1 µg/mL, rabbit anti-H4K16ac (Millipore Cat # 07-329) was used at a 1:100 dilution, rabbit anti-H3K9ac (Abcam ab10812) was used at 1 µg/mL and rabbit anti-H2AK9ac serum (Active Motif, 39109) was used at a 1:500 dilution. Donkey secondary antibodies (Jackson ImmunoResearch) were used at a final dilution of 1:400. For Figs. 3, 4 and EV1–3, Cy3 anti-rabbit (711-165-152) was used to detect H4K16ac, H3K9ac, or H2AK9ac and Cy5 anti-mouse (715-175-151) was used to detect Sirt1. For Fig. 5, Cy5 anti-rabbit (711-175-152) was used to detect H4K16ac. For Fig. EV5, Cy3 anti-rabbit was used to detect H3K9ac or H2AK9ac. For Figs. 3 and EV1–3, samples were incubated simultaneously with both primary antibodies. For Fig. 4, samples were incubated with either anti-Sirt1 or anti-H4K16ac.

## Aging regimen

For all experiments that compared aged and non-aged oocytes, mtrm$^{KG}$ smc1Δ/TM3 males (M-822) were crossed to y w virgins (A-062) to generate mtrm$^{KG}$ smc1Δ/+ virgins that were collected during an 8–12 h window and held in vials overnight with food and a small amount of dry yeast. At ~2 pm the following day, virgins were divided equally and placed in laying bottles (Fig. EV4). In one laying bottle, virgins were held in the absence of males. Because egg laying is suppressed in the absence of mating, oogenesis halts in these females and stages 8 and earlier arrest and "age" in the female (Subramanian and Bickel, 2008). Females placed into a laying bottle with X^Y, Bar (C-200) males will lay fertilized eggs continuously during the aging regimen and provide the source for "non-aged" oocytes.

Every 24 h, flies in laying bottles were provided with a fresh 60 mm petri plate containing 5% glucose/2% agar as well as a smear (~1 cm diameter) of newly prepared yeast paste (0.6 g in 1 mL sterile ultrapure water) on the agar surface. At the end of each 24-h interval, the surface of each agar plate was photographed to document egg laying (or lack thereof) for each laying bottle. Upon completion of the 4-day aging regimen, flies were transferred from the laying bottles back into vials and used for cytology or NDJ experiments. We limit our aging regimen to a 4-day time frame because virgins lay an increased number of unfertilized eggs after 4 days.

## SRT1720 supplementation

25 mM SRT1720 in DMSO (Millipore, Cat # 530748) was aliquoted upon arrival, stored at −80 °C, and a fresh aliquot was used for each experiment. On the first day of an aging regimen, one aliquot of 25 mM SRT1720 was diluted to 10 mM using a thawed aliquot of DMSO (Millipore, Cat # D2438), also stored at −80 °C. Both the 10 mM SRT1720 solution and the remaining thawed DMSO were kept at 4 °C during the 4-day aging regimen. On each day of the aging regimen, a fresh 2 mM SRT1720 solution was prepared using sterile ultrapure water. About 5 μL of 2 mM SRT1720 was added to 995 μL of sterile ultrapure water to achieve the final concentration of 10uM SRT1720, and 0.6 g of yeast was added to this solution to make yeast paste.

For the control group, we matched the concentration of DMSO in yeast paste (0.1%) to that for the SRT1720 treatment. On each day of the aging regimen, 5 μL of 100% DMSO was added to 20 μL of sterile ultrapure water to generate a 20% DMSO solution. 1 mL of 0.1% DMSO was prepared by adding 5 μL of 20% DMSO to 995 μL of sterile ultrapure water, and 0.6 g of yeast was added to this solution to generate the yeast paste.

## Age-dependent NDJ

To compare the frequency of segregation errors in aged and non-aged oocytes, $mtrm^{KG}$ $smc1\Delta/+$ females were removed from laying bottles at the end of the 4-day aging regimen and used to set up NDJ crosses in vials with $X^{\wedge}Y$, Bar (C-200) males. A smear of wet yeast (with no SRT1720 or DMSO addition) was applied to the wall of each vial. In most cases, ten vials (four females + three males) were set for each condition (aged + DMSO, non-aged + DMSO, aged + SRT1720, non-aged + SRT1720. Parents were removed after 48 h, and progeny scored through day 18.

Data from the three individual experiments are reported in Appendix Table S1. The method described in (Zeng et al, 2010) was used to determine whether aged oocytes exhibited a significant increase in NDJ compared to non-aged oocytes when mothers were fed DMSO or SRT1720. Figure 5D graphs the means from the three independent experiments. A one-way ANOVA (Statistics Kingdom), using the NDJ values from all three replicates, indicated that there was a significant difference between at least two groups ($F(3,8) = [26.8]$, $P = 0.00016$). To determine whether NDJ values differed significantly between two specific conditions, a Tukey's HSD test (Statistics Kingdom) for multiple comparisons was performed, and P values for pair-wise comparisons are presented in Fig. 5D.

## Image acquisition and processing

All images were acquired with an Andor Spinning Disk confocal (50 μm pinhole) using Nikon Elements (5.11.02 Build 1369) to control a Nikon Eclipse Ti inverted microscope, ASI MS-2000 motorized piezo stage, Zyla 4.2-megapixel sCMOS camera and three lasers (405, 561, and 637 nm). All image acquisition utilized 4X frame averaging. For all Z stack imaging, a complete Z stack was captured for each fluor sequentially, moving from longest to shortest wavelength.

For FISH imaging (ROI: 512 × 512), a CFI 100x oil Plan Apo DIC objective (NA 1.45) was used to capture a 4 μm Z series with 0.1 μm steps. For Sirt1, H4K16ac, H3K9ac, and H2AK9ac immunostaining, a Nikon CFI 40X Plan Fluor oil objective (NA 1.3) was used.

For quantification of Sirt1 and H4K16ac immunostaining on oocyte DNA (Figs. 3–5), a Z series (0.5-um step, 2 um total) was acquired for a small ROI (256 × 256) that contained the oocyte DNA. For the egg chambers shown in Figs EV1–EV3, a Z series (0.5 μm step 5 μm total) was captured for a 1024 × 1024 ROI. A small volume containing the oocyte DNA was cropped (in XY only) and used to quantify the signal of Sirt1, H4K16ac, H3K9ac, or H2AK9ac in DNA-containing voxels (Figs. EV2D and EV3D). For quantification of H3K9ac and H2AK9ac signal on oocyte DNA in aged and non-aged oocytes (Fig. EV5), a Z series (0.5 μm step, 5 μm total) was acquired for a 256 × 256 ROI.

When small ROIs were captured for quantification, an additional large image ($1024^2$ or $2048^2$) was also acquired to confirm the stage of the oocyte. Staging was based on size and morphological criteria (King, 1970; Mahowald and Kambysellis, 1980; Spradling, 1993).

For all comparisons of genotypes and/or conditions, acquisition and image processing parameters were identical for all Sirt1 images and separately for each acetylated histone. In addition, when Z series are presented, the same number of optical sections are included for each of the panels presented together in a figure.

## Quantification methods

### Scoring cohesion defects

A MATLAB (R2022a) script was used to randomize and rename FISH images so that all FISH scoring was performed blind to sample identity. Image stacks were batch processed for deconvolution (Volocity Restoration, v6.5.0) using identical parameters for all datasets. Cohesion defects were scored (Volocity Visualization, v6.5.0) by manually scrolling in all three dimensions to visualize/count the number of Cy3 and Alexa 647 probe spots on the oocyte DNA. Two spots were scored as separate if the distance between them was greater than one half the diameter of the smaller spot AND there was no evidence of a thread connecting them.

Following the scoring of both arm and centromere-proximal defects, the sample ID was revealed, and the percentage of oocytes with cohesion defects was calculated. Only arm defects were graphed in Fig. 2C, because no centromeric defects were detected in Sirt1 KD or control oocytes. A two-tailed Fisher's exact test (Graphpad) was utilized to determine whether the percentage of oocytes with arm defects were significantly different in two genotypes.

### Quantification of anti-Sirt1 and anti-H4K16ac signal on oocyte DNA

Volocity Quantitation (v6.5.0) was used for all image quantification steps. Quantification of Sirt1 and/or acetylated histone intensity on oocyte DNA was performed using the short Z series described above. The image was cropped such that the only DNA in the field was oocyte DNA (405 channel). Thresholding was used to identify the voxels containing oocyte DNA (405 nm channel). For each DNA volume, Volocity Quantification provided the average signal intensity (0-4095) for each channel: anti-Sirt1 and/or acetylated histone.

The box and whiskers plots in Figs. 3–5, EV2, EV3, and EV5 plot the average intensity of Sirt1, H4K16ac, H3K9, or H2AK9 on

DNA for each oocyte. In all box and whiskers plots shown, the average intensity is indicated with an "X", the median and quartiles are depicted by horizontal lines. Outliers are shown as solid dots and correspond to values that lie more than 1.5 * IQR (interquartile range) above the 75% or below the 25% quartile. A two-tailed unpaired *t*-test (Microsoft Excel) was used to calculate whether the difference between genotypes and/or conditions were significant. *P* values are provided on the graphs, and the number of oocytes scored is shown for each genotype.

## Data availability

This study includes no data deposited in external repositories.

The source data of this paper are collected in the following database record: biostudies:S-SCDT-10_1038-S44319-025-00634-y.

## Peer review information

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

## Acknowledgements

All Microscopy was performed in the Dartmouth Life Sciences Light Microscopy Core Facility. We thank A Lavanway and J Warren for microscopy assistance and B Johnson for fly food preparation. We are grateful to S Lacefield, MA Haseeb and D Hilpert for comments on the manuscript. We thank the Bloomington Drosophila Stock Center (NIH P40OD018537) for providing fly stocks and the Transgenic RNAi Project (NIH R24OD030002) for producing Sirt1$^{SH}$ stocks. The p4A10 anti-Sirt1 hybridoma cell line, generated by S Parkhurst (Fred Hutchinson Cancer Research Center), was obtained from the Developmental Studies Hybridoma Bank, created by NICHD of the NIH and maintained by the Department of Biology at the University of Iowa. This work was funded by NIH R01GM059354 awarded to SEB.

## Author contributions

**Zihan Meng**: Conceptualization; Formal analysis; Investigation; Visualization; Writing—original draft; Writing—review and editing. **Nicholas G Norwitz**:

Conceptualization; Formal analysis; Investigation; Writing—review and editing.
**Sharon E Bickel**: Conceptualization; Resources; Formal analysis; Supervision; Funding acquisition; Visualization; Writing—original draft; Project administration; Writing—review and editing.

Source data underlying figure panels in this paper may have individual authorship assigned. Where available, figure panel/source data authorship is listed in the following database record: biostudies:S-SCDT-10_1038-S44319-025-00634-y.

## Disclosure and competing interests statement

The authors declare no competing interests.

# Expanded View Figures

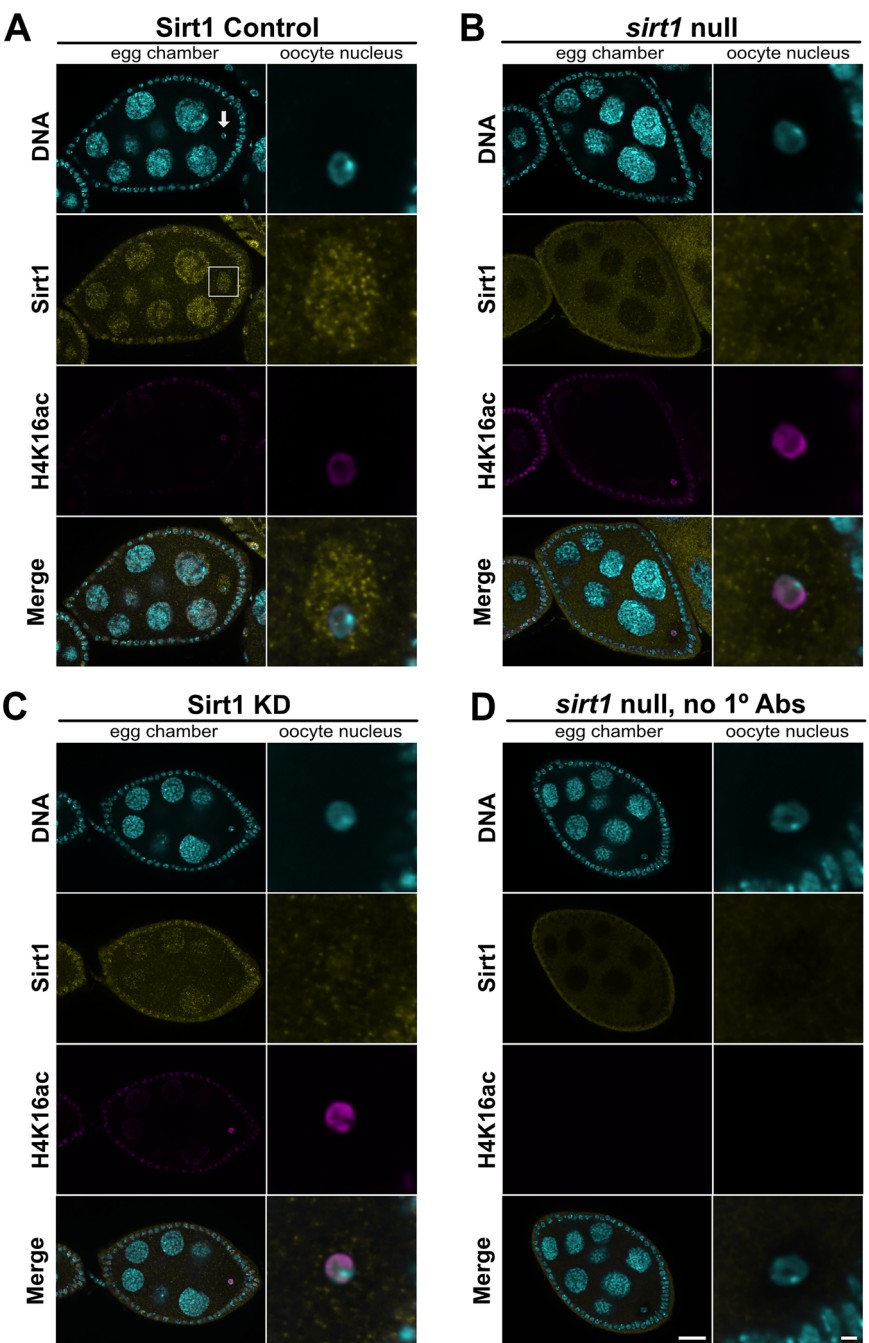

**Figure EV1. Sirt1 and H4K16ac immunolocalization in stage 7 egg chambers.**

(A–D) A single confocal optical section of a stage 7 egg chamber is shown for *Sirt1*$^{SH022-B06}$ control (no driver), KD (matα driver) and *sirt1* null. Anterior on the left. DNA (cyan), Sirt1 (yellow) and H4K16ac (magenta). Scale bars: 20 μm for egg chamber, 2 μm for oocyte nucleus. (A) Arrow points to the oocyte DNA and a white square surrounds the oocyte nucleus which is enlarged and shown to the right of each egg chamber image. (D) Primary antibodies were omitted when staining *sirt1* null ovaries. Source data are available online for this figure.

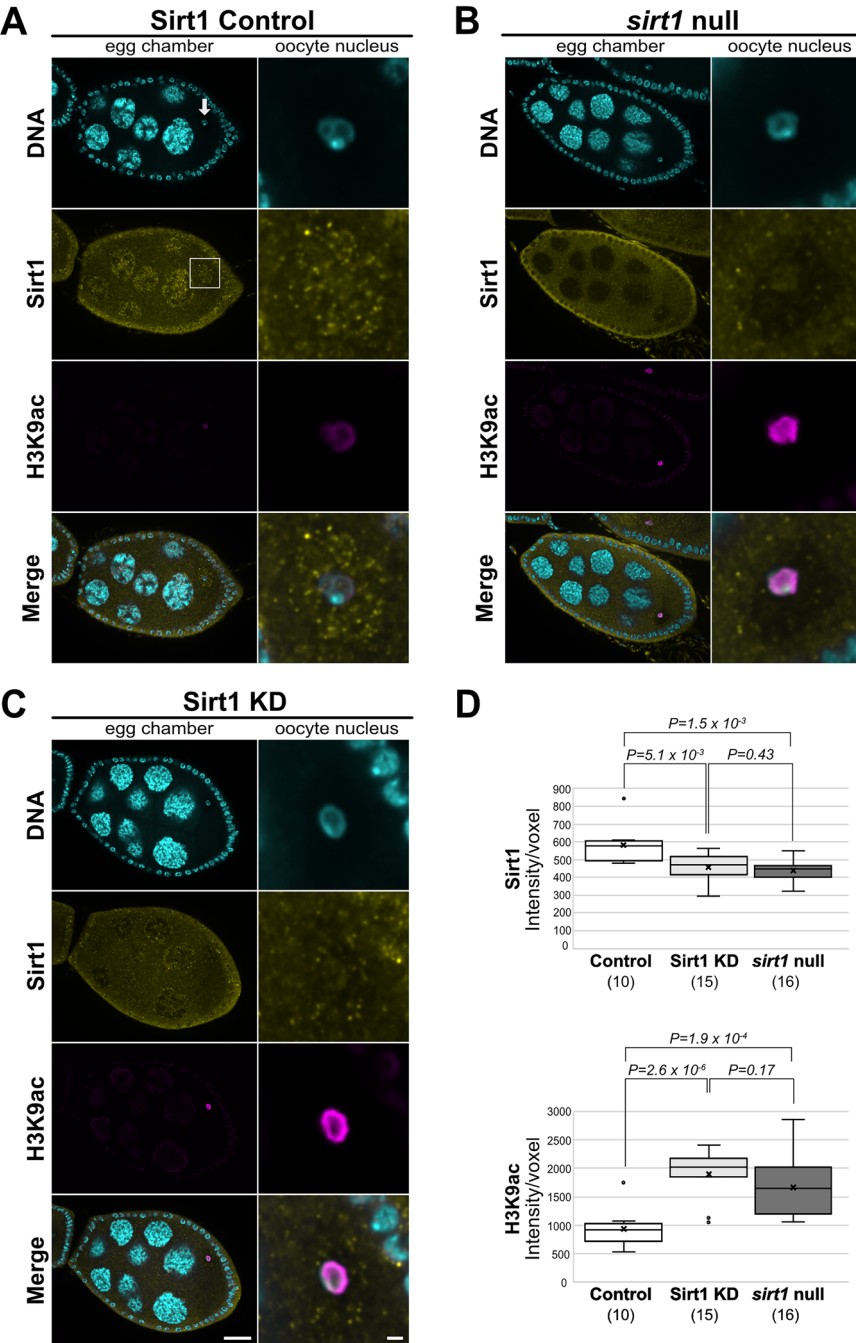

**Figure EV2. H3K9 acetylation state in stage 7 oocytes mimics that of H4K16ac.**

(A–C) A stage 7 egg chamber is shown for Sirt1^SH022-B06 control (no driver), Sirt1 KD (matα driver), and sirt1 null. Anterior on the left. DNA (cyan), Sirt1 (yellow), and H4K16ac (magenta). Single confocal optical section. Scale bars: 20 µm for egg chamber, 2 µm for oocyte nucleus. (A) An arrow indicates the oocyte DNA, and the oocyte nucleus is surrounded by a white box, which is enlarged and shown next to each egg chamber image. (D) Graphs quantify Sirt1 or H3K9ac signal colocalizing with oocyte DNA in the indicated genotypes. Number of oocytes analyzed shown in parentheses. An X marks the average with horizontal lines depicting the median and quartiles. Potential outliers are denoted with a solid black dot. Significance determined with a two-tailed t-test.

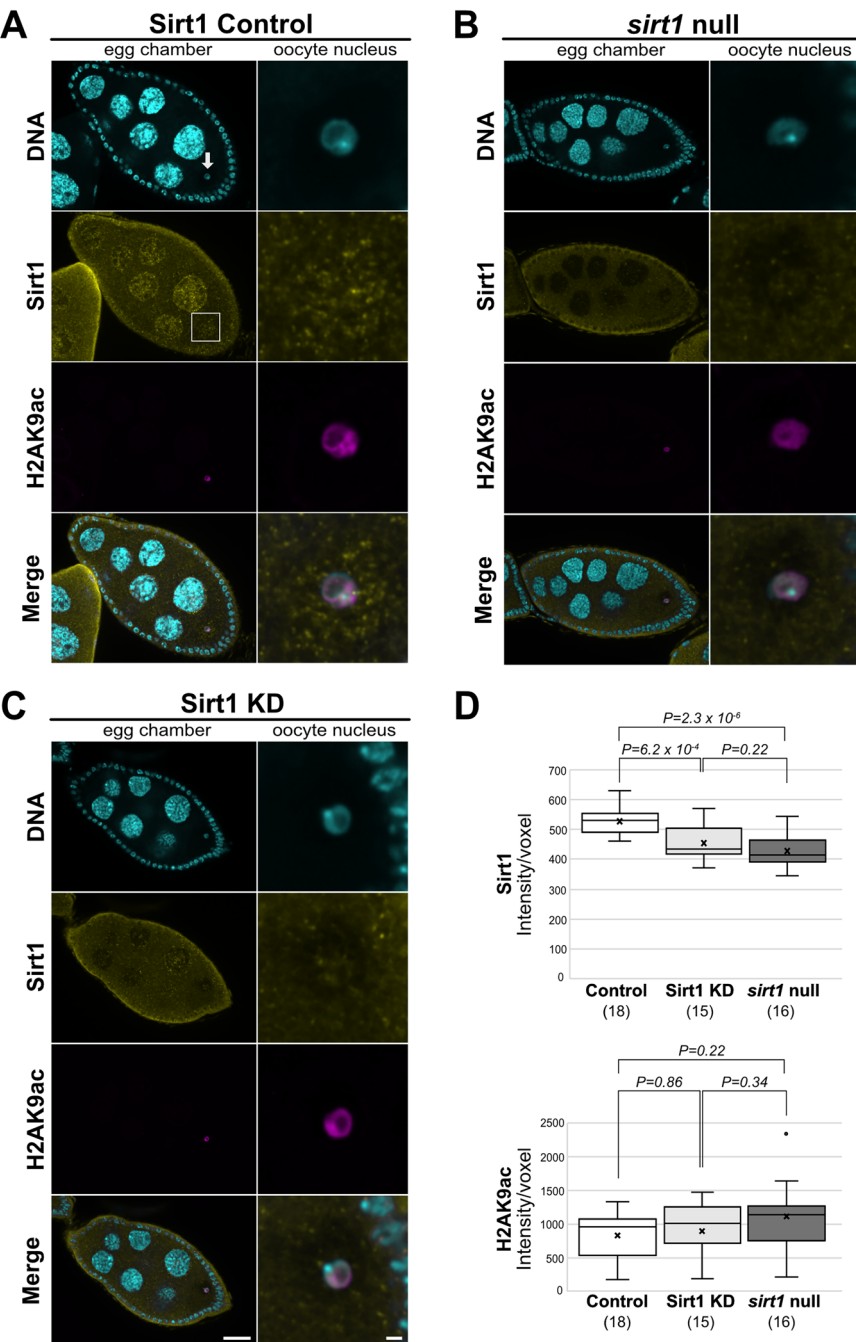

**Figure EV3.    H2AK9ac on oocyte DNA is not impacted by reduced Sirt1 activity.**

(A–C) A stage 7 egg chamber is shown for *Sirt1^SH022-B06* control (no driver), Sirt1 KD (matα driver) and *sirt1* null. Anterior on the left. DNA (cyan), Sirt1 (yellow), and H4K16ac (magenta). Single confocal optical section. Scale bars: 20 µm for egg chamber, 2 µm for oocyte nucleus. (A) The arrow points to the oocyte DNA and the oocyte nucleus is surrounded by a white box which is enlarged and shown for each egg chamber. (D) Graphs quantify Sirt1 or H2AK9ac signal colocalizing with oocyte DNA in the indicated genotypes. Number of oocytes analyzed shown in parentheses. The average is denoted with an X, and horizontal lines mark the median and quartiles. Potential outliers are denoted with a solid black dot. Significance determined with a two-tailed *t*-test.

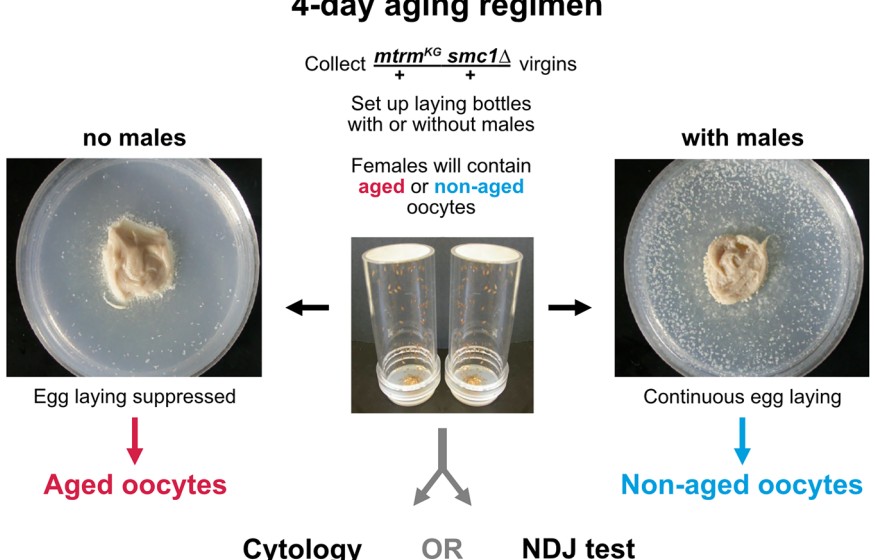

## 4-day aging regimen

Collect $\underline{mtrm^{KG}\ smc1\Delta}$ virgins
$+ \quad +$

Set up laying bottles
with or without males

Females will contain
**aged** or **non-aged**
oocytes

**no males**

Egg laying suppressed

**Aged oocytes**

**with males**

Continuous egg laying

**Non-aged oocytes**

**Cytology** OR **NDJ test**

**Figure EV4. Experimental procedure to generate and compare aged and non-aged oocytes.**

*mtrm*^KG *smc1Δ/+* virgins are placed in plastic laying bottles with a glucose/agar plate and yeast paste and *X^Y, B* males are omitted (left, aged) or added (right, non-aged). Left: In the absence of mating, egg laying is suppressed and most oogenesis stages halt progression. On the agar plate shown, very few unfertilized eggs (white spots on the agar surface) have been laid during the 24-h interval. When oogenesis halts, oocytes arrest and age at a particular stage. *Drosophila* oocytes are vulnerable to aging-induced segregation errors when they arrest and age in diplotene. Right: Oogenesis is stimulated in females that have mated, and many fertilized eggs are laid within a 24-h period. Because oogenesis is continuous in these females, they produce non-aged oocytes.

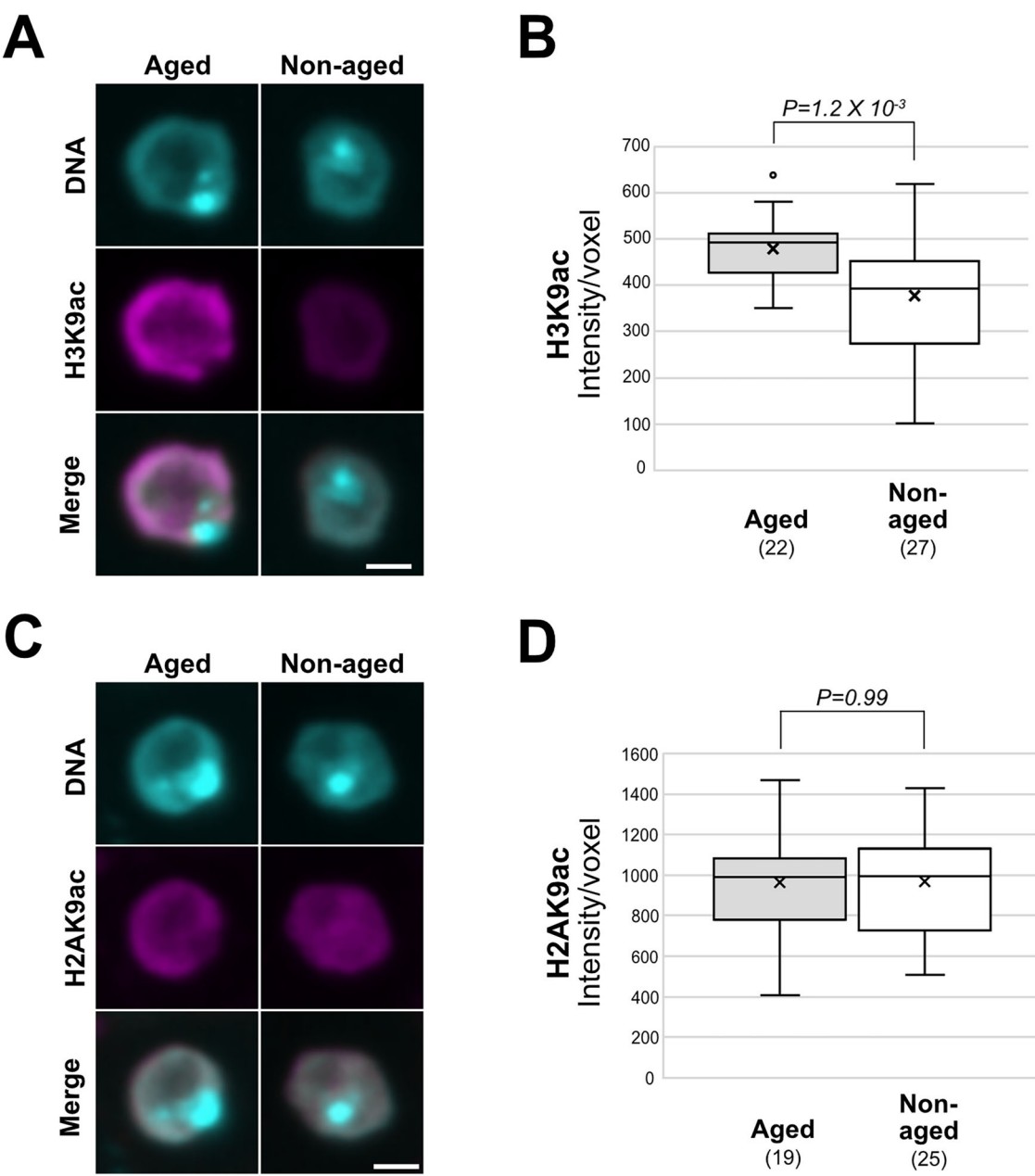

**Figure EV5.  Acetylation of H3K9 but not H2AK9 increases on oocyte DNA during aging.**

(A, C) H3K9ac or H2AK9ac immunostaining (magenta) on the DNA (cyan) of stage 7 aged and non-aged oocytes. All images are maximum intensity projections of confocal Z series. Scale bar, 2 μm. (B) Like the Sirt1 substrate H4K16ac, aging causes a significant increase in H3K9 acetylation on oocyte DNA, consistent with loss of Sirt1 deacetylase activity during aging. (D) Aging does not alter the H2AK9ac signal intensity on oocyte DNA. The number of oocytes scored in (B, D) is shown in parentheses. P values were determined using a two-tailed unpaired t-test. (B, D) An X indicates the average, with horizontal lines denoting the median and quartiles. Potential outliers are denoted with a solid black dot.

