## [Peer Review File · EMBO Reports]

Meiotic cohesion requires Sirt1 and preserving its activity in aging oocytes reduces missegregation

Sharon Bickel, Zihan Meng, and Nicholas Norwitz

Corresponding author(s): Sharon Bickel (sharon.e.bickel@dartmouth.edu)

Review Timeline:

Submission Date:	11th Mar 25
Editorial Decision:	16th Apr 25
Revision Received:	27th Jul 25
Editorial Decision:	29th Aug 25
Revision Received:	15th Oct 25
Accepted:	29th Oct 25

Editor: Deniz Senyilmaz Tiebe / Esther Schnapp

Transaction Report:

Dear Prof. Bickel,

Thank you for submitting your manuscript to EMBO Reports, which was now seen by three referees, whose reports are copied below.

Referees express interest in the proposed role of Sirt1 in age related increases in segregation errors. However, they also raise some concerns that need to be addressed to consider publication here.

I find the reports informed and constructive, and believe that addressing the concerns raised will significantly strengthen the manuscript. As the reports are below, and I think all points need to be addressed, I will not detail them here. Please contact me if you have questions or comments regarding the revision for further discussion (also by video chat).

Given these positive recommendations, we would like to invite you to revise your manuscript with the understanding that the referee concerns (as in their reports) must be fully addressed and their suggestions taken on board. Please address all referee concerns in a complete point-by-point response. Acceptance of the manuscript will depend on a positive outcome of a second round of review. It is EMBO reports policy to allow a single round of major experimental revision only and acceptance or rejection of the manuscript will therefore depend on the completeness of your responses included in the next, final version of the manuscript.

We realize that it is difficult to revise to a specific deadline. In the interest of protecting the conceptual advance provided by the work, we recommend a revision within 3 months. Please discuss the revision progress ahead of this time with me if you require more time to complete the revisions, or if you have questions or comments regarding the revision (also by video chat).

1. A data availability section providing access to data deposited in public databases is missing (where applicable).
2. Your manuscript contains statistics and error bars based on $n=2$. Please use scatter plots in these cases.

You can submit the revision either as a Scientific Report or as a Research Article. For Scientific Reports, the revised manuscript can contain up to 5 main figures and 5 Expanded View figures, and it should not exceed 27000 characters. If the revision leads to a manuscript with more than 5 main figures it will be published as a Research Article. In this case the Results and Discussion section should be separate. If a Scientific Report is submitted, these sections have to be combined. This will help to shorten the manuscript text by eliminating some redundancy that is inevitable when discussing the same experiments twice. In either case, all materials and methods should be included in the main manuscript file.

4) a .docx formatted letter INCLUDING the reviewers' reports and your detailed point-by-point responses to their comments. As part of the EMBO publication's Transparent Editorial Process, EMBO reports publishes online a Review Process File (RPF) to accompany accepted manuscripts. This File will be published in conjunction with your paper and will include the referee reports,

your point-by-point response and all pertinent correspondence relating to the manuscript.

<https://www.embopress.org/page/journal/14693178/authorguide#transparentprocess>

5) a complete author checklist, which you can download from our author guidelines

<https://www.embopress.org/page/journal/14693178/authorguide>. Please insert information in the checklist that is also reflected in the manuscript. The completed author checklist will also be part of the RPF.

6) Please note that all corresponding authors are required to supply an ORCID ID for their name upon submission of a revised manuscript (<<https://orcid.org/>>). Please find instructions on how to link your ORCID ID to your account in our manuscript tracking system in our Author guidelines

<<https://www.embopress.org/page/journal/14693178/authorguide#authorshipguidelines>>

7) Before submitting your revision, primary datasets produced in this study need to be deposited in an appropriate public database (see <https://www.embopress.org/page/journal/14693178/authorguide#datadeposition>). Please remember to provide a reviewer password if the datasets are not yet public. The accession numbers and database should be listed in a formal "Data Availability" section placed after Materials & Method (see also

<https://www.embopress.org/page/journal/14693178/authorguide#datadeposition>). Please note that the Data Availability Section is restricted to new primary data that are part of this study. * Note - All links should resolve to a page where the data can be accessed. *

Additional information on source data and instruction on how to label the files are available:

<https://www.embopress.org/page/journal/14693178/authorguide#sourcedata>

9) Our journal encourages inclusion of *data citations in the reference list* to directly cite datasets that were re-used and obtained from public databases. Data citations in the article text are distinct from normal bibliographical citations and should directly link to the database records from which the data can be accessed. In the main text, data citations are formatted as follows: "Data ref: Smith et al, 2001" or "Data ref: NCBI Sequence Read Archive PRJNA342805, 2017". In the Reference list, data citations must be labeled with "[DATASET]". A data reference must provide the database name, accession number/identifiers and a resolvable link to the landing page from which the data can be accessed at the end of the reference. Further instructions are available at <http://www.embopress.org/page/journal/14693178/authorguide#referencesformat>

10) Regarding data quantification (see Figure Legends:

<https://www.embopress.org/page/journal/14693178/authorguide#figureformat>)

- the name of the statistical test used to generate error bars and P values,

- the number (n) of independent experiments (please specify technical or biological replicates) underlying each data point,

- the nature of the bars and error bars (s.d., s.e.m.),

- If the data are obtained from n Program fragment delivered error ``Can't locate object method "less" via package "than" (perhaps you forgot to load "than"?) at //ejpvfs23/sites23b/embor_www/letters/embor_decision_revise_and_review.txt line 56.' 2, use scatter blots showing the individual data points.

12) Please also note our reference format:

13) All Materials and Methods need to be described in the main text using our 'Structured Methods' format, which is required for all research articles. According to this format, the Methods section includes a Reagents and Tools Table (listing key reagents, experimental models, software and relevant equipment and including their sources and relevant identifiers) followed by a Methods and Protocols section describing the methods using a step-by-step protocol format. The aim is to facilitate adoption of the methodologies across labs. More information on how to adhere to this format as well as a downloadable template (.docx) for the Reagents and Tools Table can be found in our author guidelines:

I look forward to seeing a revised version of your manuscript when it is ready. Please let me know if you have questions or comments regarding the revision.

Kind regards,

Deniz Senyilmaz Tiebe

Deniz Senyilmaz Tiebe, PhD
Senior Scientific Editor
EMBO Reports

Referee #1:

In this manuscript, Zihan et al. show that Sirt1 controls meiotic cohesion in *Drosophila* oocytes via its deacetylase activity, and that preserving Sirt1 activity reduces chromosome missegregation in aging oocytes. Sirt1 depletion in the female germline heterozygous for *mtrmKG/+* (which exhibits weakened meiotic cohesion) leads to increased segregation errors. This depletion is associated with elevated levels of H4K16 acetylation. Interestingly, in aged oocytes (from 4-day-old unmated *mtrmKGsmc1Δ/+* females), acetylated H4K16 levels are higher compared to non-aged oocytes (from 4-day-old mated *mtrmKGsmc1Δ/+* females), suggesting that Sirt1 activity is suppressed in aged oocytes. Furthermore, feeding flies with the Sirt1 activator SRT1720 reduces H4K16 acetylation and decreases chromosome segregation errors in aged oocytes. While the findings are interesting, several concerns remain and should be addressed prior to publication.

Major points:

1. The authors knocked down Sirt1 in the *mtrmKG/+* background and observed 8-10 % NDJ (Fig.1). Have the authors also knocked down Sirt1 in a wildtype background? What % of NDJ is present in that case? If the result is negative or shows only a very low NDJ frequency, does that suggest that Sirt1 plays only a minor function in meiotic cohesion? In addition, NDJ should be spelled out as nondisjunction when it first appears in the text. The function of *mtrm* should also be explained in more detail for clarity. Furthermore, since there is sufficient space in the main figures, it would be helpful for reader accessibility to move Figure EV1 into the main figure panel. This would make it easier for readers outside the field to follow experimental design and logic.
2. Knockdown of Sirt1 caused arm cohesion defects in about 30% of oocytes that are wildtype for *mtrm* (Fig. 2). It is unclear why the authors did not perform these experiments in the *mtrmKG/+* background, given their focus on sensitized genetic condition. Further, if 30% cohesion defects are already observed upon Sirt1 knockdown in a wild-type germline, it raises the question of why the NDJ assay required introduction of a *mtrm* mutant allele. Clarifying this point would help readers better understand the rationale behind the experimental design and interpretation of the data.
3. Although the authors used oocytes from non-mating *mtrmKGsmc1Δ/+* females as aged oocytes (Fig. EV2 and Fig. 4 and 5), it will be more convincing if the author also examined Sirt1 and H4K16ac expression in wildtype oocytes with age, tested whether treatment of aged wild-type flies with a Sirt1 activator could similarly reduce cohesion defects. Additionally, it would be helpful if the authors could provide a clearer explanation of *smc* in the text, particularly for readers less familiar with the cohesion machinery.
4. It is informative to know if Sirt1 activity is specifically decreased in aged oocytes or also other ovarian cells (nurse cells and follicle cells).
5. Sirt1 and Sirt2 are known to be involved in the regulation of cohesion during meiosis. In addition to findings from yeast

studies, relevant research in other systems should be discussed or summarized in the Discussion section to help readers better appreciate the significance of the advanced findings presented in this manuscript.

Minor points:

1. The term "pericentric cohesion" should be briefly explained when it first appears in the text to aid reader understanding.
2. In Table EV1, the Bickel stock numbers may not be meaningful to readers outside of Dr. Bickel's lab.

Referee #2:

The manuscript by Meng et al. addresses an important question: How is meiotic cohesion preserved during oocyte aging? Human oocytes are paused during meiosis from birth until ovulation. As oocytes age one of the factors that contributes to increased aneuploidy is a loss of sister chromatid cohesion.

Here the authors examine the role of a NAD⁺-dependent deacetylase, Sirt1, in maintaining sister chromatid cohesion in *Drosophila* aged oocytes. *Drosophila* do not have the same meiotic pause as humans, but the authors use a system they previously developed to recreate a meiotic pause that results in aged oocytes. They use RNAi to show that a reduction of Sirt1 results in an increase in meiotic non-disjunction and increased H4K16 acetylation. Further, they show that aging results in an increase of acetylation. However, Sirt1 levels do not change with age. They suggest that the increase in acetylation is due to a loss of Sirt1 function. This is a nicely written manuscript however, there are some things that can be done to improve it.

The major comment is that the authors say multiple times that the increase in acetylation in aged oocytes is due to a loss of Sirt1 function. There is not enough evidence to support this claim in my opinion. Sirt1 levels are not changed in aged oocytes and there are no experiments addressing Sirt1 function. The evidence that a Sirt1 activator decreases acetylation in aged oocytes does point in the direction of Sirt1 activity being important, but it could be that an increase in Sirt1 activity is compensating for the loss of a different deacetylase. For example: Are there other deacetylases that target H4K16? Are other Sirt1 targets gaining acetylation? Post translational modifications and their modifiers are complex and without more direct evidence this claim is not supported. While the discussion does address other possibilities the authors need to temper the language used in the abstract/results sections.

For the activator experiment- it would have been nice to have a control of sirt1 RNAi knockdown flies that were fed the activator. This would show that the activator is truly only activating Sirt1 activity in flies and the deacetylation is not an off-target effect.

Data in figure 3 and 4 with the Sirt1 antibody is not very convincing. I agree it is less in the null, but it also looks very non-specific due to the zoomed in images. It would be more convincing if the authors showed a less zoomed in image (more similar to the supplement image that was more convincing). Further, a no primary control in the supplement would be useful.

Line 78-79: was this from this paper? Otherwise needs a citation. It was not clear if this was past data or this paper.

Figures: In general, the figures were lacking in the labeling present on the figure itself. The information is present in the legends but for easy of reading it would be helpful to have more information on the figures. For example:

Figure 1: KD of what? NDJ of what chromosome?

Figure 2:

- Panel B should be labelled with what the colors are and what the different panels represent.
- KD of what?

Figure 3: Labels on graphs- KD of what? Null genotype? Both graphs should be labelled for the genotypes.

Figure 5: Panel A is unnecessary, and that information is in the supplement.

Table EV2: NDJ of what chromosome?

Referee #3:

Here the authors use sophisticated genetic manipulation and a chemical Sirt1 activator, SRT1720, to study the role of Sirt1 during ageing in *Drosophila* oocytes. They provide evidence that oocyte chromosome mis-segregation increases when Sirt1 is compromised. Using an ageing model, they provide evidence that Sirt1 activity declines with ageing, albeit not Sirt1 protein

levels. Finally, they show that SRT1720 ameliorates the ageing effect on chromosome mis-segregation. These are interesting findings that address an area of great interest pertaining to age-related decline in oocyte quality and more broadly, a relationship between Sirt1 and cohesion. They provide a potential therapeutic option for averting age-related oocyte quality decline. However, I have a number of concerns.

Comments:

1. There appears no direct evidence of Sirt1 knockdown using the Gal4/UAS strategy. Can the authors provide evidence of this akin to that used to show Sirt1 depletion in Sirt1 nulls in EV3.
2. The title of the paper is "Meiotic cohesion requires Sirt1...." and throughout the paper the authors refer to studying cohesion. Significantly, however, there is no direct evaluation of cohesin proteins with the only evidence for a "cohesion defect" being FISH-based analysis of the separation of DNA probes. One of the sections is titled "Cohesion maintenance in prophase oocytes depends on Sirt1", however, the data they present (whilst inferential), really shows a link between Sirt1 and sister chromatid proximity. How is it known that this separation is due to cohesion breakdown rather than another factor, for instance, altered chromatin structure? For a paper focused squarely on "cohesion", there should be much more direct evidence regarding cohesin (e.g. immunostaining).
3. The authors do not find that age-related increase in H4K16ac signals (used as a readout for Sirt1 deacetylase activity) is due to a decline in Sirt1 protein levels. Declining Sirt1 activity with age is pivotal to this paper's message but relies on a single measure, H4K16ac signals. Is there an additional marker to corroborate that Sirt1 activity does decline with age? Why then does Sirt1 activity decline if protein levels remain stable? Do the levels of its essential co-activator, NAD, decline with age as has been shown for mouse oocytes (Bertolo et al 2020 - not cited here)?
4. It is notable that oocyte-specific loss of Sirt1 in mouse oocytes has no impact on the integrity of chromosome segregation in either young or aged mouse oocytes albeit there is a later effect on preimplantation embryo development (Iljas et al. 2020). Notably, as well, loss of two other key sirtuins, Sirt2 and Sirt3, have no discernible impact on mouse oocytes (Bertoldo et al. 2020; Iljas & Homer 2020). Can the authors reconcile this difference between flies and mice?
5. The authors used the Sirt1 null model to validate their antibody but not for functional studies of chromosome mis-segregation. Does mis-segregation similarly occur at high rates in this model as in Sirt1 KD?
6. The authors find that treatment with SRT1720 sustains low H4K16ac signals with ageing suggesting that it promotes Sirt1 activity. Moreover, this is associated with suppressed chromosome mis-segregation with ageing suggesting that Sirt1 activation could ameliorate a major consequence of oocyte ageing. However, chemical activators are notoriously problematic. There doesn't appear to be much validation of this activator in this experimental system, for instance, dose-finding studies. How did the authors settle on 10 μ M? Is there a dose-dependent effect? Can the authors replicate this with an independent Sirt1 activator? If decline of Sirt1 activity with ageing is not due to reduced Sirt1 protein, presumably it may well be due to age-related NAD decline. If lack of Sirt1's co-activator, NAD, is indeed limiting, how can SRT1720 augment Sirt1 activity? These are important points to clarify if Sirt1 activators are to be considered legitimate interventions for safeguarding oocyte quality in the clinical sphere.
7. The authors infer a relationship between Sirt1 and cohesion but provide no mechanistic insight into how this might come about. The authors published a paper showing that oxidative stress causes loss of cohesion in Drosophila oocytes (Perkins et al. 2016). Given the well-known antioxidant properties of Sirt1, this would seem the logical first step to investigate. Do ROS levels increase during ageing in Drosophila oocytes and in Sirt1 KD and null oocytes? If so, does SRT1720 combat high ROS? The mechanism is important since Sirt1 could be important for either preventing cohesin degradation with age or promoting cohesin loading. This is a fairly straightforward question to resolve using SRT1720 and other Sirt1 activators. If it turns out that Sirt1 promotes cohesin loading in Drosophila oocytes - which the authors recently showed turn over cohesin during prophase (Haseeb et al. 2024) - how would this relate to mammalian oocytes which appear not to turnover cohesin during prophase-arrest (Tachibana-Konwolski et al. 2010).

Overall, this is a potentially interesting study. However, the results are very preliminary and considerably more validation and mechanistic insight are required.

EMBOR-2025-61501V1
Point-by-point Responses to Reviewer Comments

- *Please find our bulleted responses to individual comments in blue italics.*
- *We thank all three reviewers for their questions, comments and suggestions, all of which have helped us to improve the manuscript.*

Reviewer 1:

In this manuscript, Zihan et al. show that Sirt1 controls meiotic cohesion in *Drosophila* oocytes via its deacetylase activity, and that preserving Sirt1 activity reduces chromosome missegregation in aging oocytes. Sirt1 depletion in the female germline heterozygous for *mtrm*^{KG/+} (which exhibits weakened meiotic cohesion) leads to increased segregation errors. This depletion is associated with elevated levels of H4K16 acetylation. Interestingly, in aged oocytes (from 4-day-old unmated *mtrm*^{KGsmc1Δ/+} females), acetylated H4K16 levels are higher compared to non-aged oocytes (from 4-day-old mated *mtrm*^{KGsmc1Δ/+} females), suggesting that Sirt1 activity is suppressed in aged oocytes. Furthermore, feeding flies with the Sirt1 activator SRT1720 reduces H4K16 acetylation and decreases chromosome segregation errors in aged oocytes. While the findings are interesting, several concerns remain and should be addressed prior to publication.

1. The authors knocked down Sirt1 in the *mtrm*^{KG/+} background and observed 8-10 % NDJ (Fig.1). Have the authors also knocked down Sirt1 in a wildtype background? What % of NDJ is present in that case? If the result is negative or shows only a very low NDJ frequency, does that suggest that Sirt1 plays only a minor function in meiotic cohesion?
 - *We have now included the results for a NDJ test in *mtrm*⁺ oocytes using the Sirt1 SH00806.N hairpin: Control → 0.27% and KD → 1.30%, P value = 0.43 (lines 114-117). However, we do not think this means that Sirt1 plays a minor function in cohesion because the FISH experiments (Fig 2C) were performed with *mtrm*⁺ oocytes and >30% of the Sirt1 KD oocytes exhibited arm cohesion defects, strongly supporting a prominent role for Sirt1 in cohesion maintenance.*
 - *To help readers better understand the reason we measure X-chromosome NDJ in *mtrm*^{KG/+} heterozygotes, we have now provided a new visual in Fig 1 (panel C) that illustrates how the *Drosophila* oocyte achiasmate system affects our NDJ assay. In *Drosophila*, we need to disable this oocyte backup system so that bivalents that have lost arm cohesion will missegregate (in addition to achiasmate chromosomes). Importantly, both Sirt1 KD and control oocytes are heterozygous for the *mtrm*^{KG} allele, so the difference in NDJ between these two genotypes is due to a reduction in Sirt1 in KD oocytes.*

In addition, NDJ should be spelled out as nondisjunction when it first appears in the text. The function of *mtrm* should also be explained in more detail for clarity. Furthermore, since there is sufficient space in the main figures, it would be helpful for reader accessibility to move Figure EV1 into the main figure panel. This would make it easier for readers outside the field to follow experimental design and logic.

- *Nondisjunction is now spelled out on line 100.*
- *On lines 102-3, we also state that we are using the term NDJ broadly to include any type of chromosome segregation error.*

- *The original **Fig EV1** has been eliminated, and its contents have been moved to **Figs 1 & 2**.*
2. Knockdown of Sirt1 caused arm cohesion defects in about 30% of oocytes that are wildtype for mtrm (Fig. 2). It is unclear why the authors did not perform these experiments in the mtrm^{KG/+} background, given their focus on sensitized genetic condition. Further, if 30% cohesion defects are already observed upon Sirt1 knockdown in a wild-type germline, it raises the question of why the NDJ assay required introduction of a mtrm mutant allele. Clarifying this point would help readers better understand the rationale behind the experimental design and interpretation of the data.
 - *As indicated in the response to comment #1, we hope that **Fig 1C (new visual)** helps readers to understand why the mtrm^{KG/+} sensitized genetic background is only required for NDJ test and not for visualizing cohesion defects using FISH. In FISH experiments, we are not relying on missegregation as a way to detect loss of cohesion. Instead, we are looking at the state of cohesion directly. In addition, because the mtrm^{KG/+} genotype also has a minor effect on arm cohesion (Bonner et al, 2020; Haseeb et al, 2024b), we do not want this mutation present when we assay cohesion defects in Sirt1 KD oocytes. Because the Sirt1 KD oocytes are mtrm+ in **Figure 2C**, we can conclude that cohesion defects result from the reduction of Sirt1 protein caused by knockdown during meiotic prophase.*
 3. Although the authors used oocytes from non-mating mtrm^{KG}smc1 Δ /+ females as aged oocytes (Fig. EV2 and Fig. 4 and 5), it will be more convincing if the author also examined Sirt 1 and H4K16ac expression in wildtype oocytes with age, tested whether treatment of aged wild-type flies with a Sirt1 activator could similarly reduce cohesion defects. Additionally, it would be helpful if the authors could provide a clearer explanation of smc in the text, particularly for readers less familiar with the cohesion machinery.
 - *Our aging experiments are restricted to the mtrm^{KG} smc1 Δ /+ genotype because in our prior work, we observed that wild-type Drosophila oocytes (mtrm+ smc1+) that undergo our 4-day aging regimen do not exhibit age-dependent NDJ (Subramanian & Bickel, 2008). In these oocytes, a functional achiasmate system ensures accurate segregation of recombinant bivalents that lose arm cohesion. In addition, the 4-day aging regimen does not increase NDJ when oocytes have normal levels of the cohesin subunit, SMC1. We cannot extend the aging regimen beyond four days because egg laying starts to increase, which means that fewer oocytes arrest and age. However, in our prior work we found that with the two-fold reduction of functional cohesin in smc1 Δ /+ oocytes, four days of oocyte aging is sufficient to cause a significant increase in NDJ due to loss of cohesion (Subramanian & Bickel, 2008).*
 - *We have revised the paragraph describing the genotype we use for aging experiments to provide a clearer rationale for why we need to use mtrm^{KG} smc1 Δ /+ flies for our aging regimen and age-dependent NDJ tests. In the same paragraph, we also introduce SMC1 as a cohesin subunit (**lines 219-225**).*
 4. It is informative to know if Sirt1 activity is specifically decreased in aged oocytes or also other ovarian cells (nurse cells and follicle cells).
 - *Interestingly, we and others (Samata et al, 2020) have observed that the H4K16ac signal is readily visible on oocyte DNA, but the signal is very weak or absent on nurse cell DNA and follicle cell DNA (as shown in **Fig EV1A**). Even though H4K16ac signal on oocyte DNA is approximately two-fold greater in sirt1 null oocytes (**Fig 3C**), we still do not observe H4K16ac signal on nurse cell DNA in the null. Although the H4K16ac*

*nuclear signal in the follicle cells appears to increase in some sirt1 null egg chambers (such as the one in **Fig EV1B**), we do not consistently observe this increase.*

- Our new experiments indicate that the egg chamber staining pattern for H3K9ac, another known Sirt1 substrate, is similar to that of H4K16ac (see **Fig EV2**).*
5. Sirt1 and Sirt2 are known to be involved in the regulation of cohesion during meiosis. In addition to findings from yeast studies, relevant research in other systems should be discussed or summarized in the Discussion section to help readers better appreciate the significance of the advanced findings presented in this manuscript.
- We are confused by the above statement. Despite numerous searches, we have not found any literature that links Sirt1 or Sir2 with cohesion in meiotic cells.*
 - We now additionally mention two papers that connect Sirt1 or Sirt2 with cohesin.*
 - However, we believe that our results are novel and provide the first link between Sirt1 and sister chromatid cohesion in oocytes.*

Minor points:

1. The term "pericentric cohesion" should be briefly explained when it first appears in the text to aid reader understanding.

- We have omitted the term "pericentric" throughout the text and used terms such as "centromere-proximal" or "heterochromatin near the centromere".*

2. In Table EV1, the Bickel stock numbers may not be meaningful to readers outside of Dr. Bickel's lab.

- While we acknowledge that this may be the case, we have decided to keep this info in the table because we use it in our cross descriptions in the Methods and it makes it easier for researchers to request specific stocks. However, because of the addition of **new EV Figures**, this info is now provided in **Appendix Table S2**.*

Referee #2:

The manuscript by Meng et al. addresses an important question: How is meiotic cohesion preserved during oocyte aging? Human oocytes are paused during meiosis from birth until ovulation. As oocytes age one of the factors that contributes to increased aneuploidy is a loss of sister chromatid cohesion.

Here the authors examine the role of a NAD⁺-dependent deacetylase, Sirt1, in maintaining sister chromatid cohesion in Drosophila aged oocytes. Drosophila do not have the same meiotic pause as humans, but the authors use a system they previously developed to recreate a meiotic pause that results in aged oocytes. They use RNAi to show that a reduction of Sirt1 results in an increase in meiotic non-disjunction and increased H4K16 acetylation. Further, they show that aging results in an increase of acetylation. However, Sirt1 levels do not change with age. They suggest that the increase in acetylation is due to a loss of Sirt1 function. This is a nicely written manuscript however, there are some things that can be done to improve it.

The major comment is that the authors say multiple times that the increase in acetylation in aged oocytes is due to a loss of Sirt1 function. There is not enough evidence to support this claim in my opinion. Sirt1 levels are not changed in aged oocytes and there are no experiments addressing Sirt1 function. The evidence that a Sirt1 activator decreases acetylation in aged

oocytes does point in the direction of Sirt1 activity being important, but it could be that an increase in Sirt1 activity is compensating for the loss of a different deacetylase.

For example: Are there other deacetylases that target H4K16? Are other Sirt1 targets gaining acetylation? Post translational modifications and their modifiers are complex and without more direct evidence this claim is not supported. While the discussion does address other possibilities the authors need to temper the language used in the abstract/results sections.

- *We have performed additional experiments to address concerns about using the acetylation state of H4K16 as the sole marker for Sirt1 enzymatic activity/function in vivo.*
- *H3K9ac is another well-established substrate of Sirt1. Using an antibody specific for H3K9ac, we now show that acetylation of H3K9 on oocyte DNA increases significantly in sirt1 null oocytes (Fig EV2) and also when oocytes undergo aging (Fig EV5). This phenotype mimics that of H4K16ac.*
- *Based on the literature, H2AK9ac is not a substrate for Sirt1 deacetylation. Consistent with this, acetylation of H2AK9 on oocyte DNA does not change in sirt1 null oocytes or when oocytes undergo aging (Fig EV3 & EV5).*
- *Given our findings that oocyte aging causes two known substrates of Sirt1 to become hyperacetylated but does not impact the acetylation state of a Sirt1 “non-substrate,” we think our data provide strong evidence that Sirt1 activity declines during aging; however, we have also softened our language throughout the text, including section headings and figure legend titles.*

For the activator experiment- it would have been nice to have a control of sirt1 RNAi knockdown flies that were fed the activator. This would show that the activator is truly only activating Sirt1 activity in flies and the deacetylation is not an off-target effect.

- *We agree. However, for reasons we do not understand, we were unable to generate the necessary flies to perform this experiment in a sirt1 null background.*
- *For Sirt1 KD flies, we have previously found that un-mated females containing the strong *mat α* driver chromosome do not hold their eggs during our 4-day aging regimen and, as a result, we cannot examine the effect of oocyte aging in this genotype. We have successfully used a weak *mat α* driver for oocyte aging experiments (Perkins et al, 2019), but unfortunately this would not provide sufficient knockdown of Sirt1 for the proposed experiment.*

Data in figure 3 and 4 with the Sirt1 antibody is not very convincing. I agree it is less in the null, but it also looks very non-specific due to the zoomed in images. It would be more convincing if the authors showed a less zoomed in image (more similar to the supplement image that was more convincing). Further, a no primary control in the supplement would be useful.

- *Using images from new experiments, we now include enlarged images of the oocyte nucleus for each whole egg chamber image we present (please see Figs EV1, EV2 and EV3). Enrichment of Sirt1 in the oocyte nucleus is visible for control oocytes, but not Sirt1 KD or sirt1 null oocytes. Note, the nucleus is much larger than the oocyte DNA mass.*
- *In addition, we have also added a “no primary antibody” control to Fig EV1. The Sirt1 channel in this control (Fig EV1D) has a pattern that is similar, but dimmer, than the cytoplasmic signal that observe in the sirt1 null egg chambers (Fig EV1B). These data suggest that the residual Sirt1 signal in sirt1 null (Fig EV1B) is due to non-specific binding of the anti-Sirt1 antibody.*

- *We cannot display larger areas for Figures 3-5, because the image stacks captured for quantification were small ROIs, and we want to show images that were part of each quantification set.*
- *Although we agree that the Sirt1 signal looks relatively diffuse when zooming in on the oocyte DNA (Figs 3-4), we are confident regarding our quantification of Sirt1 “associated with DNA”. For each 3D image stack, we measured Sirt1 signal intensity only for DNA-containing voxels and the Sirt1 signal associated with oocyte DNA is significantly higher for control oocytes than for Sirt1KD or sirt1 null oocytes.*

Line 78-79: was this from this paper? Otherwise needs a citation. It was not clear if this was past data or this paper.

- *We have changed the beginning of this sentence to read “We show here....” to make it clear that we are summarizing results that are presented in our manuscript (line 76)*

Figures: In general, the figures were lacking in the labeling present on the figure itself. The information is present in the legends but for easy of reading it would be helpful to have more information on the figures. For example:

Figure 1: KD of what? NDJ of what chromosome?

Figure 2:

-Panel B should be labelled with what the colors are and what the different panels represent.

-KD of what?

Figure 3: Labels on graphs- KD of what? Null genotype? Both graphs should be labelled for the genotypes.

Figure 5: Panel A is unnecessary, and that information is in the supplement.

Table EV2: NDJ of what chromosome?

- *For Figs 1, 2 & 3 and Appendix Table S1 (formerly Table EV2), we have added the requested information.*
- *Panel A in Fig 5 provides information about our SRT1720 experimental design that is not included in what is now Fig EV4, so we think it should remain in Fig 5.*

Referee #3:

Here the authors use sophisticated genetic manipulation and a chemical Sirt1 activator, SRT1720, to study the role of Sirt1 during ageing in *Drosophila* oocytes. They provide evidence that oocyte chromosome mis-segregation increases when Sirt1 is compromised. Using an ageing model, they provide evidence that Sirt1 activity declines with ageing, albeit not Sirt1 protein levels. Finally, they show that SRT1720 ameliorates the ageing effect on chromosome mis-segregation.

These are interesting findings that address an area of great interest pertaining to age-related decline in oocyte quality and more broadly, a relationship between Sirt1 and cohesion. They provide a potential therapeutic option for averting age-related oocyte quality decline. However, I have a number of concerns.

Comments:

1. There appears no direct evidence of Sirt1 knockdown using the Gal4/UAS strategy. Can the authors provide evidence of this akin to that used to show Sirt1 depletion in Sirt1 nulls in EV3.

- **Fig 3** (which has not changed) provides quantitative evidence that Sirt1 signal on oocyte DNA in Sirt1 KD oocytes is significantly lower than in Sirt1 control oocytes. However, these data are restricted to the oocyte DNA.
- We now include stage 7 whole egg chamber images for Sirt1 KD in **Figs EV1, EV2 and EV3** to provide visual confirmation of knock down for this genotype. Although Sirt1 signal was significantly reduced in the oocyte nuclei of Sirt1 KD females, residual signal persists in the polyploid nurse cells. Also note that because the *mat α* driver is germline specific, Sirt1 signal in the follicle cells surrounding the egg chamber is not affected by KD.

2. The title of the paper is "Meiotic cohesion requires Sirt1...." and throughout the paper the authors refer to studying cohesion. Significantly, however, there is no direct evaluation of cohesin proteins with the only evidence for a "cohesion defect" being FISH-based analysis of the separation of DNA probes. One of the sections is titled "Cohesion maintenance in prophase oocytes depends on Sirt1", however, the data they present (whilst inferential), really shows a link between Sirt1 and sister chromatid proximity. How is it known that this separation is due to cohesion breakdown rather than another factor, for instance, altered chromatin structure? For a paper focused squarely on "cohesion", there should be much more direct evidence regarding cohesin (e.g. immunostaining).

- *Using these same probes and FISH assay, we have previously demonstrated that *mat α* -induced knockdown of the cohesin subunit, SMC3, or the cohesin loader, Nipped-B, each cause a significant increase in the number of oocytes with arm cohesion defects (3-4 arm spots)(Haseeb et al, 2024a; Haseeb et al., 2024b) . Therefore, we are confident that separated spots do represent loss of cohesion. We have added this information, including citations, to the text on **lines 135-139**.*
- *In addition, we have moved the results of our recombinational history assay to **Fig 2E**, so they follow the FISH results. Our finding that Sirt1KD causes recombinant homologs to missegregate at increased frequency confirms that the separated arm signals in our FISH assay are functionally relevant and do reflect loss of arm cohesion.*

3. The authors do not find that age-related increase in H4K16ac signals (used as a readout for Sirt1 deacetylase activity) is due to a decline in Sirt1 protein levels. Declining Sirt1 activity with age is pivotal to this paper's message but relies on a single measure, H4K16ac signals. Is there an additional marker to corroborate that Sirt1 activity does decline with age? Why then does Sirt1 activity decline if protein levels remain stable? Do the levels of its essential co-activator, NAD, decline with age as has been shown for mouse oocytes (Bertolo et al 2020 - not cited here)?

- *As mentioned above, we have examined the acetylation state of H3K9, another Sirt1 substrate, and found that like H4K16, acetylation of H3K9 on oocyte chromosomes increases significantly in *sirt1* null oocytes and with aging (**Fig EV2**).*
- *In addition, an acetylation mark (H2AK9ac) that is not dependent on Sirt1 deacetylation does not change in *sirt1* null oocytes or with age (**Fig EV3**).*
- *We have not tried to measure NAD⁺ levels in *Drosophila* oocytes. A biochemical approach to measure NAD⁺ in ovary extracts will not be useful because the lysates would contain material from both somatic and germline cells. In addition, we do not have access to a microscope that allows multispectral imaging to measure NAD⁺ optically, as performed by Bertoldo et al. (2020).*
- *We now cite the Bertoldo paper (Bertoldo et al, 2020) when discussing NMN supplementation (**line 326**).*

4. It is notable that oocyte-specific loss of Sirt1 in mouse oocytes has no impact on the integrity of chromosome segregation in either young or aged mouse oocytes albeit there is a later effect on preimplantation embryo development (Iljas et al. 2020). Notably, as well, loss of two other key sirtuins, Sirt2 and Sirt3, have no discernible impact on mouse oocytes (Bertoldo et al. 2020; Iljas & Homer 2020). Can the authors reconcile this difference between flies and mice?

- *The tools available to disrupt protein function and analyze outcomes in mouse oocytes and Drosophila oocytes are quite different and may contribute to the differences between mouse and fly results. The Iljas et al. paper (Iljas et al, 2020) utilized live analysis to monitor chromosome segregation but did not assay for sister chromatid cohesion defects or measure chromosome segregation errors directly. Although oocyte specific knockout of Sirt1 did not cause overt problems with bipolar spindle formation or anaphase onset in mouse oocytes, the timing between GVBD and chromosome alignment was modestly, but significantly, delayed.*
- *In our experiments, the $mat\alpha$ driver is strong enough for robust knockdown but is not expressed until mid-prophase, leaving early meiotic events unaffected. This tool has been pivotal for our work (Haseeb et al., 2024a; Haseeb et al., 2024b; Perkins et al, 2016; Perkins et al., 2019; Weng et al, 2014). In addition, our genetic assay allows us to sample thousands of oocyte meiotic divisions and reproducibly detect missegregation of the X chromosome. A test like this is not feasible in mice.*
- *In addition, there may be redundancy in mouse oocytes that is missing in Drosophila oocytes, making it easier to uncover a phenotype caused by loss of Sirt1 function in flies. There are seven sirtuins in mammals, but only five in Drosophila. Mammalian Sirt2 localizes to nuclei and the cytosol and shares substrates with Sirt1. In contrast, Drosophila Sirt2, which shares homology with both mammalian Sirt2 and Sirt3, has been reported to function within mitochondria, like mammalian Sirt3 (Rahman et al, 2014), making it unlikely that Sirt1 and Sirt2 have substrates in common in Drosophila.*
- *Although Matrimony is not conserved beyond Drosophila species, evidence of a backup system that promotes accurate segregation of achiasmate chromosomes has also been reported for yeast meiotic cells and mouse spermatocytes (Kurdzo & Dawson, 2015). All three achiasmate systems rely on centromeric associations between homologs. Moreover, an estimate of the frequency at which human chromosome 21 fails to achieve a crossover and the frequency at which it missegregates led to the proposal that a backup system may also operate in human oocytes (Koehler & Hassold, 1998). Like the effect of the Drosophila oocyte achiasmate system on our NDJ test, existence of such a system in mouse oocytes would promote accurate segregation of bivalents even if arm cohesion distal to the crossover were lost due to oocyte-specific knockout of mouse Sirt1.*
- *We have now included a paragraph in the discussion that summarizes the points above.*

6. The authors used the Sirt1 null model to validate their antibody but not for functional studies of chromosome mis-segregation. Does mis-segregation similarly occur at high rates in this model as in Sirt1 KD?

- *As mentioned above, we were unable to generate the flies necessary to measure NDJ in $sirt1$ null; $mtrm^{KG}/+$ oocytes (see response to Reviewer 2)*
- *While deletion knockouts are excellent tools, in this particular case, we think that prophase KD of Sirt1 in $mtrm^{KG}/+$ oocytes (**Fig 1**) is actually the cleanest way to perform this experiment. In contrast to the $sirt1$ null, $mat\alpha$ -induced KD is germline specific and the $mat\alpha$ driver does not turn on until after the pachytene checkpoint, a process in which Sirt1 has been implicated (Joyce & McKim, 2010). Thus, NDJ results in the $sirt1$ null would likely be confounded by disruption of the pachytene checkpoint.*

- *We think that the NDJ data we present for two different Sirt1 hairpins (Fig 1E) provide strong functional evidence that Sirt1 functions during meiotic prophase to promote accurate segregation of meiotic chromosomes in Drosophila oocytes.*
7. The authors find that treatment with SRT1720 sustains low H4K16ac signals with ageing suggesting that it promotes Sirt1 activity. Moreover, this is associated with suppressed chromosome mis-segregation with ageing suggesting that Sirt1 activation could ameliorate a major consequence of oocyte ageing. However, chemical activators are notoriously problematic. There doesn't appear to be much validation of this activator in this experimental system, for instance, dose-finding studies. How did the authors settle on 10 μ M? Is there a dose-dependent effect? Can the authors replicate this with an independent Sirt1 activator? If decline of Sirt1 activity with ageing is not due to reduced Sirt1 protein, presumably it may well be due to age-related NAD decline. If lack of Sirt1's co-activator, NAD, is indeed limiting, how can SRT1720 augment Sirt1 activity? These are important points to clarify if Sirt1 activators are to be considered legitimate interventions for safeguarding oocyte quality in the clinical sphere.
- *Our understanding is that the initial controversy (Pacholec et al, 2010) regarding the mechanism of small molecule Sirt1 activators has been resolved (Dai et al, 2015; Dai et al, 2010).*
 - *We chose SRT1720 for our experiments because it was the most potent activator available from a well-established vendor.*
 - *We chose to use 10 μ M because this is the highest concentration for which Sirt1 activity has been measured in vitro (Milne et al, 2007).*
 - *Unlike mouse studies, which typically add SRT1720 to drinking water, our experiments must deliver the activator in food (there is no separate water source). Therefore, we mixed SRT1720 into a paste made with water and live yeast that flies eat during the aging regimen. We chose to use a high concentration of SRT1720 in the paste because it is unclear how much activator is taken up (and potentially broken down) by metabolically active yeast.*
 - *We did not test a different activator; our survey of recent literature indicates that testing a second activator in addition to SRT1720 is quite rare.*
 - *We think our finding that consumption of SRT1720 by mothers decreases acetylation of the Sirt1 substrate H4K16ac in aging oocytes (as would be expected for a Sirt1 activator) provides strong validation of this activator in our experimental system.*
 - *Structural studies indicate that the allosteric effect of Sirt1 small molecule activators increase affinity for NAD⁺ as well as its substrate. This would explain how SRT1720 supplementation can augment Sirt1 activity even if NAD⁺ is limiting (Dai et al., 2015; Dai et al., 2010).*
8. The authors infer a relationship between Sirt1 and cohesion but provide no mechanistic insight into how this might come about. The authors published a paper showing that oxidative stress causes loss of cohesion in Drosophila oocytes (Perkins et al. 2016). Given the well-known antioxidant properties of Sirt1, this would seem the logical first step to investigate. Do ROS levels increase during ageing in Drosophila oocytes and in Sirt1 KD and null oocytes? If so, does SRT1720 combat high ROS? The mechanism is important since Sirt1 could be important for either preventing cohesin degradation with age or promoting cohesin loading. This is a fairly straightforward question to resolve using SRT1720 and other Sirt1 activators. If it turns out that Sirt1 promotes cohesin loading in Drosophila oocytes - which the authors recently showed turn over cohesin during prophase

(Haseeb et al. 2024) - how would this relate to mammalian oocytes which appear not to turnover cohesin during prophase-arrest (Tachibana-Konwolski et al. 2010).

- *These are interesting questions to pursue that will hopefully provide insight into the mechanism by which Sirt1 promotes meiotic cohesion. We discuss some of these ideas near the end of the paper (in both the original submission and the revised manuscript). However, we believe this additional experimentation lies beyond the scope of our present study.*

Overall, this is a potentially interesting study. However, the results are very preliminary and considerably more validation and mechanistic insight are required.

- *We think that our work represents a carefully executed quantitative analysis that has uncovered a novel role for Sirt1 in oocytes and provides clear evidence that feeding mothers a Sirt1 activator while their oocytes age can suppress age-dependent segregation errors.*
- *As such, we think our story is well suited for the journal's description of a **Scientific Report**: "conceptually striking, self-contained observations based on compelling data."*

REFERENCES CITED

Bertoldo MJ, Listijono DR, Ho WJ, Riepsamen AH, Goss DM, Richani D, Jin XL, Mahbub S, Campbell JM, Habibalahi A et al (2020) NAD(+) Repletion Rescues Female Fertility during Reproductive Aging. Cell Rep 30: 1670-1681 e1677

Bonner AM, Hughes SE, Hawley RS (2020) Regulation of Polo Kinase by Matrimony Is Required for Cohesin Maintenance during Drosophila melanogaster Female Meiosis. Curr Biol 30: 715-722 e713

Dai H, Case AW, Riera TV, Considine T, Lee JE, Hamuro Y, Zhao H, Jiang Y, Sweitzer SM, Pietrak B et al (2015) Crystallographic structure of a small molecule SIRT1 activator-enzyme complex. Nat Commun 6: 7645

Dai H, Kustigian L, Carney D, Case A, Considine T, Hubbard BP, Perni RB, Riera TV, Szczepankiewicz B, Vlasuk GP et al (2010) SIRT1 activation by small molecules: kinetic and biophysical evidence for direct interaction of enzyme and activator. J Biol Chem 285: 32695-32703

Haseeb MA, Bernys AC, Dickert EE, Bickel SE (2024a) An RNAi screen to identify proteins required for cohesion rejuvenation during meiotic prophase in Drosophila oocytes. G3 (Bethesda) 14

Haseeb MA, Weng KA, Bickel SE (2024b) Chromatin-associated cohesin turns over extensively and forms new cohesive linkages in Drosophila oocytes during meiotic prophase. Curr Biol 34: 2868-2879 e2866

Ilijas JD, Wei Z, Homer HA (2020) Sirt1 sustains female fertility by slowing age-related decline in oocyte quality required for post-fertilization embryo development. Aging Cell 19: e13204

Koehler KE, Hassold TJ (1998) Human aneuploidy: lessons from achiasmate segregation in Drosophila melanogaster. Ann Hum Genet 62: 467-479

Kurdzo EL, Dawson DS (2015) Centromere pairing--tethering partner chromosomes in meiosis I. *FEBS J* 282: 2458-2470

Milne JC, Lambert PD, Schenk S, Carney DP, Smith JJ, Gagne DJ, Jin L, Boss O, Perni RB, Vu CB et al (2007) Small molecule activators of SIRT1 as therapeutics for the treatment of type 2 diabetes. *Nature* 450: 712-716

Pacholec M, Bleasdale JE, Chrunyk B, Cunningham D, Flynn D, Garofalo RS, Griffith D, Griffor M, Loulakis P, Pabst B et al (2010) SRT1720, SRT2183, SRT1460, and resveratrol are not direct activators of SIRT1. *J Biol Chem* 285: 8340-8351

Perkins AT, Das TM, Panzera LC, Bickel SE (2016) Oxidative stress in oocytes during midprophase induces premature loss of cohesion and chromosome segregation errors. *Proc Natl Acad Sci U S A* 113: E6823-E6830

Perkins AT, Greig MM, Sontakke AA, Peloquin AS, McPeck MA, Bickel SE (2019) Increased levels of superoxide dismutase suppress meiotic segregation errors in aging oocytes. *Chromosoma* 128: 215-222

Rahman M, Nirala NK, Singh A, Zhu LJ, Taguchi K, Bamba T, Fukusaki E, Shaw LM, Lambright DG, Acharya JK et al (2014) *Drosophila* Sirt2/mammalian SIRT3 deacetylates ATP synthase beta and regulates complex V activity. *J Cell Biol* 206: 289-305

Samata M, Alexiadis A, Richard G, Georgiev P, Nuebler J, Kulkarni T, Renschler G, Basilicata MF, Zenk FL, Shvedunova M et al (2020) Intergenerationally Maintained Histone H4 Lysine 16 Acetylation Is Instructive for Future Gene Activation. *Cell* 182: 127-144 e123

Subramanian VV, Bickel SE (2008) Aging predisposes oocytes to meiotic nondisjunction when the cohesin subunit SMC1 is reduced. *PLoS Genet* 4: e1000263

Weng KA, Jeffreys CA, Bickel SE (2014) Rejuvenation of Meiotic Cohesion in Oocytes during Prophase I Is Required for Chiasma Maintenance and Accurate Chromosome Segregation. *PLoS genetics* 10: e1004607

Dear Prof. Bickel,

Thank you for submitting your revised manuscript. It has now been seen by two of the original referees.

As you will see, referees find that the study is significantly improved during revision and recommend publication. However, the editorial points below need to be addressed before I can accept the manuscript.

- Please place the Disclosure And Competing Interests Statement after the Acknowledgements.
- Please remove the Author Contributions section from the manuscript text.
- The Funding section should be included in the Acknowledgements section and the title Funding should be removed.
- We note the following regarding figure callouts: "Appendix Table 2" in the reagents table is missing "S"
- We note the following regarding the Appendix table: Please add page numbers in the Table of Contents (first page) and please correct the nomenclature in all places (callouts included) - i.e. Appendix Table S1, Appendix Table S2.
- Please remove the Reagents & Tools table from the manuscript text and upload it separately in word format by choosing the appropriate file type.
- Please resubmit the source data as one zip file per figure.
- Please rename the Methods and Protocols section as Methods.
- During our routine figure checks we note the following:
 - o The same photo of fly vials was potentially reused in left and right parts of Figure 5A and Figure EV4. Please confirm that this photo was used for representational purposes only.
 - o Figure EV1D has two empty cells in HK416ac - i.e. we fail to detect any background signal. Was any signal detected for these conditions? Please provide source data for this panel.
- Our production/data editors have asked you to clarify several points in the figure legends - Figure Legends (main + EV):
 - o Please note that the exact p values are not provided in the legends of figures 1E, 2E
 - o Please indicate the statistical test used for data analysis in the legends of figures 1E, 3B, C
 - o Please note that the box plots need to be defined in terms of minima, maxima, centre, bounds of box and whiskers, and percentile in the legends of figures 3B, C; 4B, D; 5C, EV2 D, EV3 D, EV5B, D
- Papers published in EMBO Reports include a 'synopsis' and 'bullet points' to further enhance discoverability. Both are displayed on the html version of the paper and are freely accessible to all readers. The synopsis includes a short standfirst summarizing the study in 1 or 2 sentences (max 35 words) that summarize the paper and are provided by the authors and streamlined by the handling editor. I would therefore ask you to include your synopsis blurb and 3-5 bullet points listing the key experimental findings.
- In addition, please provide an image for the synopsis. This image should provide a rapid overview of the question addressed in the study but still needs to be kept fairly modest since the image size cannot exceed 550 (width) x 300-600 (height) pixels.

Thank you again for giving us to consider your manuscript for EMBO Reports, I look forward to your minor revision.

Kind regards,

Deniz Senyilmaz Tiebe

--

Deniz Senyilmaz Tiebe, PhD
Senior Scientific Editor
EMBO Reports

Referee #1:

The current manuscript is easy to read. The authors have addressed all my concerns.

Referee #2:

The authors have addressed all of my concerns and the manuscript is improved. I appreciate the authors thoroughness in their responses. This version is suitable for publication in my opinion.

All editorial and formatting issues were resolved by the authors.

Prof. Sharon Bickel
Dartmouth College
Biological Sciences
78 College Street
Hanover, NH 03755
United States

Dear Prof. Bickel,

I am very pleased to accept your manuscript for publication in the next available issue of EMBO reports. Thank you for your contribution to our journal.

Yours sincerely,
